# TOPOSCORER: A LIGHT, INTERPRETABLE PREDICTOR FOR PROTEIN-PROTEIN BINDING AFFINITY

## ABSTRACT

Protein-protein binding affinity underlies complex stability, selectivity, and therapeutic action, yet experimental measurement is low-throughput. Although a number of deep learning models are now end-to-end differentiable, they generally lack interpretable attributions, whereas traditional topology-based affinity predictors rely on non-differentiable persistent diagrams or barcodes. We present TopoScorer, a lightweight, interpretable, end-to-end–trainable affinity scorer that can act as a loss or reward to steer generative and discriminative protein models; across protein and mutation affinity benchmarks, it delivers performance comparable to state-of-the-art methods and, when integrated into a modern antibody-design workflow, improves affinity-related metrics of generated candidates. The core component of TopoScorer is Specter(Spectral Topology Encoder), a topology-driven, multi-channel, multi-scale differentiable feature extractor for protein–protein interfaces that converts full-atom coordinates into topo-spectral representations via Persistent Topological Hyperdigraph Laplacians (PTHLs) and differentiable spectral descriptors, preserving physicochemical-role–aware cues alongside 3D topological structure to yield compact, interpretable features suitable for learning.

## 1 INTRODUCTION

Protein–protein binding affinity underlies complex stability and selectivity in immune recognition and drug action. It is a key quantity for understanding pathway regulation, predicting mutation effects, and guiding molecular design. Experimental assays such as isothermal titration calorimetry (Velazquez-Campoy & Freire, 2006) and surface plasmon resonance(Rich & Myszka, 2000) measure thermodynamic or kinetic parameters directly, but they are low-throughput and costly, limiting coverage of the large sequence and conformational spaces needed for modern design–validation cycles. Physics-based energy models—including force-field scoring, MM/GBSA (Genheden & Ryde, 2015), FEP (Wang et al., 2015), and docking rescoring (Trott & Olson, 2010)—implemented in mature toolchains such as PyRosetta(Chaudhury et al., 2010) and FoldX (Delgado et al., 2019; Schymkowitz et al., 2005) are interpretable, but they are sensitive to sampling and parameter choices, and expensive to run at scale.

Deep learning has become the dominant paradigm for protein–protein binding affinity prediction, delivering state-of-the-art accuracy and throughput(Cai et al., 2024)(Jin et al., 2023)(Shan et al., 2022; Yu et al., 2024; Yue et al., 2025)(Luo et al., 2023). Learning-based methods span supervised structural models (Luo et al., 2023; Jiménez et al., 2018; Li et al., 2021), unsupervised energy-shaped models (Jin et al., 2023), flow-based models (Luo et al., 2023), surface based models(Mallet et al., 2025; Song et al., 2024; Banerjee et al., 2025) and sequence language models(Meier et al., 2021b; Hsu et al., 2022a). Despite their strong empirical performance, most models still face limitations common to deep learning in structural biology, including limited interpretability of the learned features and high training costs at scale(Luo et al., 2023)(Cai et al., 2024). Moreover, mainstream protein generation and design models simply do not optimize binding affinity—affinity is absent from their objectives, so no gradient signal is available to shape the formation of high-affinity complexes.

To address these gaps, TopoScorer—a lightweight, interpretable, end-to-end-trainable affinity scorer that predicts binding affinity directly from full-atom protein coordinates, enabling end-to-

end training while preserving interpretability. TopoScorer is built on Specter, a topology-driven, multichannel-encoded, differentiable PPI feature extractor, and a lightweight predictor network. Seeking interpretability rather than another black-box scorer, we anchor the method in the determinants of affinity—namely, the interface's three-dimensional geometry and its heterogeneous physicochemical environment (electrostatics, hydrophobicity, hydrogen bonding, steric effects). A useful representation must therefore capture both the 3D topological structures and multi-channel physicochemical cues in a unified form, which is non-trivial in practice. Persistant Topological Hyperdigraph Laplacians(PTHLs)(Chen et al., 2024; 2023) is an effective topological representation that excel at capturing multiscale topological features of 3D structures, yet their non-differentiability impedes end-to-end learning. In this work, we applies multi-scale soft filtrations to extract cross-protein PTHLs and further obtain topological spectral features with differentiable spectral descriptors (e.g.,Laplacian eigenvalue statistics). To capture both geometric and chemical cues at the interface, we use a multi-channel design that preserves physicochemical-role–aware features—charge, hydrophobicity, and hydrogen-bond donor/acceptor patterns, leading to a general topological spectral features extractor that transforms protein–protein interfaces into multi-scale, multi-channel features suitable for downstream machine-learning models of affinity.

We validate TopoScorer on proteins and mutations affinity benchmarks and integrate it into a state-of-the-art antibody design workflow, where it significantly improves affinity-related metrics of the generated candidates. In summary, Our research makes the following contributions:

- TopoScorer, a light, interpretable PPI binding-affinity predictor built on multi-channel, multi-scale topological–spectral features.

- Specter(Spectral Topology Encoder), a differentiable feature extractor for protein-protein interfaces, encoding both structural and physicochemical information.

- We present, to our knowledge, the first demonstration that a differentiable deep-learning affinity predictor can directly steer a generative antibody design model, improving sequence–structure co-design and supplying a reliable, scalable training signal that drives the generation of higher-affinity complexes.

## 2 RELATED WORK

### 2.1 PROTEIN-PROTEIN BINDING AFFINITY PREDICTION

Accurate prediction of protein-protein binding (PPB) affinity is critical for screening protein therapeutics. Traditional interfacial contact analysis and surface property calculations are time-consuming and lack of accuracy. Classical structure-based baselines and modern deep models span a coherent spectrum. PRODIGY(Xue et al., 2016) estimates PPI affinity from interfacial contacts and surface descriptors and remains a strong structural reference. DeepSite(Jiménez et al., 2017) localizes pockets on voxelized protein maps, while KDEEP(Jiménez et al., 2018) and subsequent work(Stepniewska-Dziubińska et al., 2018) validate grid CNNs for affinity prediction on PDBbind. Moving beyond voxels, atomistic GNNs(Li et al., 2021) encode distances, angles, and pairwise relations to capture nonlocal interactions. Complementing supervised models, DSMBind(Jin et al., 2023) learns SE(3)-equivariant generative energy signals in an unsupervised manner results across various PPI tasks. As for protein mutation effects, DDGPred(Shan et al., 2022; Yu et al., 2024; Yue et al., 2025) provides an end-to-end framework for $\Delta\Delta G$ and is widely used for mutational ranking in antibody optimization; GearBind(Cai et al., 2024) is a pretrainable geometric graph neural network for protein-protein binding affinity change prediction; Pi-SAGE(Banerjee et al., 2025) is a permutation-invariant, surface-aware graph encoder that learns residue-level surface tokens from protein structure and augments an all-atom GNN like GearBind with explicit, context-aware surface features; in parallel, RDE-Network leverages side-chain rotamer density to capture conformational entropy/flexibility and transfer to PPI $\Delta\Delta G$ prediction (Luo et al., 2023). Pre-trained protein language models such as ESM-1v (Meier et al., 2021a) and ESM-IF (Hsu et al., 2022b) can be finetuned for affinity prediction, but their training on monomeric sequences limits their ability to capture information from critical "hotspot" residues that largely determine binding. Concurrently, advances in structure prediction—exemplified by AlphaFold-Multimer (Evans et al., 2021b), AlphaFold 3 (Abramson et al., 2024), and Boltz-2 (Passaro et al., 2025)—have improved complex modeling; their confidence scores are often used as rough proxies for binding affinity. Notably, among these

methods only Boltz-2 is designed to predict an affinity score directly, and it currently supports only limited-length ligands. For interpretability, DeepAffinity proposed a sequence-based representation for affinity (Karimi et al., 2019), and ANTIPASTI(Michalewicz et al., 2024) delivers an interpretable antibody–antigen affinity predictor, but not directly applicable to generic PPIs. Reviews of protein structural modeling and design underscore the need for physically grounded, interpretable models that integrate structure with machine learning to predict generic PPIs (Gao et al., 2020; Casadio et al., 2022; Omar et al., 2023; Ding et al., 2022).

## 2.2 TOPOLOGICAL DEEP LEARNING

Early work in Topological Data Analysis(TDA) established filtrations and persistence to track the birth–death of homology classes, represented by barcodes diagrams and persistence diagrams(Carlsson, 2009), providing robust global structure and broad cross-modal applicability and remain resilient to noise. Topological deep learning integrates algebraic-topological structure into neural models so they can represent and learn higher-order, multiscale, shape-aware patterns beyond pairwise relations, yielding strong performance across diverse tasks(Som et al., 2018; Reininghaus et al., 2015; Singh et al., 2008). TopologyNet (Cang & Wei, 2017) pioneered the use of element-specific persistent homology (ESPH) to extract multichannel topological signatures from biomolecular structure and feed them to CNNs for property/affinity prediction. TopoNetTree(Cang & Wei, 2017) is a classic persistent-homology model that combines ESPH with CNNs, which starts from topological representations and relies on Betti-number–based persistent homology barcodes as features. Chen et al.(Chen et al., 2023) introduce persistent hyperdigraph homology and the persistent hyperdigraph Laplacians (PTHLs). TopoFormer(Chen et al., 2024) integrates PTHLs-derived, element-specific multiscale topological sequences with a Transformer encoder, converting 3D protein–ligand structures into NLP-admissible tokens and achieving strong structure-to-sequence prediction performance on protein-ligand docking, screening and scoring tasks. Our differentiable topological–spectral features build on this line by enabling gradient flow from spectral/topological statistics back to coordinates, and we further introduce a PPI-specific channel encoding scheme that organizes interfacial atoms/physicochemical roles to better capture recognition-relevant geometry.

## 3 METHODS

### 3.1 DIFFERENTIABLE TOPOLOGICAL FEATURES EXTRACTION

**Hypergraph-Induced Cross-Protein Distance Matrices**    Given a protein–protein complex with proteins $A$ and $B$, we denote the heavy atom coordinates by $X^A = \{x_u\}_{u \in \mathcal{V}^A}$ and $X^B = \{x_v\}_{v \in \mathcal{V}^B}$. We partition $X^A$ and $X^B$ into physicochemical-role–aware classes $G_1, \ldots, G_N$(Table 6) and, within each channel, designate as putative interface atoms those whose minimum distance to any atom in the opposite protein falls below a cutoff $r_c$.

For a cross-chain channel pair $(i, j)$ with $i \in \mathcal{C}^A$ and $j \in \mathcal{C}^B$, the vertex set is $\mathcal{V}_i^A \uplus \mathcal{V}_j^B$ with coordinates $X^A = \{x_u\}_{u \in \mathcal{V}^A}$ and $X^B = \{x_v\}_{v \in \mathcal{V}^B}$. A (directed) hypergraph $\mathcal{H} = (\mathcal{V}, \mathcal{E})$ consists of vertices $\mathcal{V}$ and oriented $k$-hyperedges $e = (v_0 \rightarrow \cdots \rightarrow v_k)$ $(k \geq 1)$, i.e., ordered $(k+1)$-tuples of distinct vertices; reversing order flips orientation. We quantify geometry via a cross-protein distance function:

$$D^{AB} \in \mathbb{R}^{|\mathcal{V}_i^A| \times |\mathcal{V}_j^B|}, \quad D_{uv}^{AB} = \|x_u - x_v\|_2,$$

and define a unified pairwise distance

$$d(p, q) = \begin{cases} \|x_p - x_q\|_2, & p \in \mathcal{V}_i^A,\ q \in \mathcal{V}_j^B \text{ or } p \in \mathcal{V}_j^B,\ q \in \mathcal{V}_i^A, \\ \bar{d}, & \text{otherwise}, \end{cases}$$

where $\bar{d} := \max_{u \in \mathcal{V}_i^A,\, v \in \mathcal{V}_j^B} D_{uv}^{AB} + \varepsilon$ (a large finite constant) so that intra-protein pairs are effectively "far."

**Soft Filtration to Persistent Topological Hyperdigraph Laplacians**    Over radii $\{r_t\}_{t=1}^T$ (distance filtration), replace hard cut-offs with the smooth gate

$$\kappa_\tau(d; r_t) = \sigma\!\left(\frac{r_t - d}{\tau}\right), \qquad \sigma(z) = \frac{1}{1 + e^{-z}},\ \ \tau > 0,$$

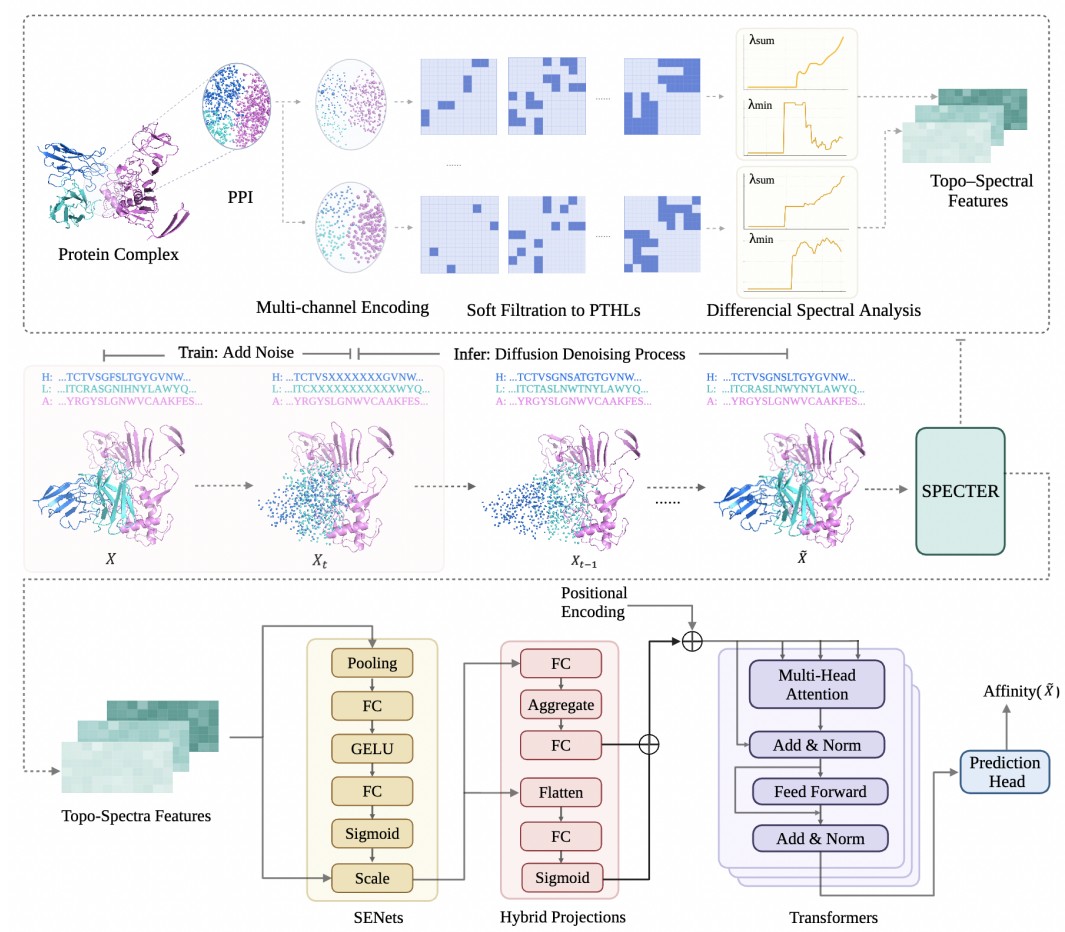

Figure 1: Overview of differentiable topological spectral feature extractor Specter in TopoScorer, diffusion model finetuning pipeline and structure of binding affinity predictor.

which is differentiable in the coordinates and converges to a hard threshold as $\tau \to 0^+$. At each scale $r_t$, build a weighted directed hypergraph $\mathcal{H}^{(i,j)}(r_t)$ on the disjoint union $\mathcal{V}_i^A \uplus \mathcal{V}_j^B$. For an oriented $k$-hyperedge $e = (v_0 \to \cdots \to v_k)$, assign a weight using only cross-protein pairs,

$$w_e(r_t) = \prod_{\{p,q\} \in \mathcal{X}(e)} \kappa_\tau(d(p,q); r_t), \qquad \mathcal{X}(e) = \{\{p,q\} \subset \{v_0, \ldots, v_k\} : p \in \mathcal{V}_i^A, \ q \in \mathcal{V}_j^B\}.$$

so that intra-protein proximities are excluded by construction. Let $C_k(r_t)$ be the real vector space of $k$-chains (formal sums of oriented $k$-hyperedges) and define the signed boundary map on generators by

$$\partial_k(r_t)(v_0 \to \cdots \to v_k) = \sum_{m=0}^{k} (-1)^m (v_0 \to \cdots \widehat{v_m} \cdots \to v_k),$$

where $\widehat{v_m}$ denotes omission with the induced orientation. Equip $C_k(r_t)$ with the weight-aware inner product $\langle e, e' \rangle_k = e^T W_k(r_t) e'$ with $W_k(r_t) = \mathrm{Diag}\{w_e(r_t)\}$ and write $B_k$ for the signed incidence matrix of $\partial_k(r_t)$. With these inner products, the matrix of the Hilbert adjoint $\partial_k^*(r_t)$ is

$$\partial_k^*(r_t) = W_k(r_t)^{-1} B_k^\top W_{k-1}(r_t).$$

The $k$-th Hodge Laplacian on chains is

$$L_k^{(i,j)}(r_t) = \partial_k^*(r_t) \partial_k(r_t) + \partial_{k+1}(r_t) \partial_{k+1}^*(r_t) = W_k^{-1} B_k^\top W_{k-1} B_k + B_{k+1} W_{k+1}^{-1} B_{k+1}^\top W_k,$$

which is self-adjoint with respect to $\langle \cdot, \cdot \rangle_k$ and positive semidefinite. Its kernel encodes $k$-order harmonic (topological) features, while the positive spectrum captures geometric organization at scale $r_t$. Varying $t$ yields a differentiable, multi-scale family $\{L_k^{(i,j)}(r_t)\}$ that summarizes cross-chain interaction topology under a soft filtration.

**Differentiable topo–spectral features.** Given a symmetric positive semidefinite Laplacian $L \in \mathbb{R}^{n \times n}$, we summarize its topology with a six-tuple of differentiable spectral statistics of its eigenvalues. We only differentiate through eigenvalues (not eigenvectors) and use a small diagonal shift $\varepsilon I$ with a fallback schedule to handle near-singular/ill-conditioned cases. We then form several Laplacian eigenvalue statistics as spectral features.

A smooth surrogate of $\max(\lambda_i)$ by log-sum-exp is $\lambda_{\max}^{\mathrm{soft}} = \tau \log \sum_i \exp(\lambda_i/\tau)$, with $\tau > 0$.

Sum of eigenvalues equals to trace of the Laplacian, that is, $\lambda_{\mathrm{sum}}^{\mathrm{soft}} = \mathrm{tr}(L_s)$.

Zero count is approximated by counting "near-zero" eigenvalues via a Gaussian kernel centered at $0$ with bandwidth $\sigma_0 > 0$, that is, $\lambda_{\mathrm{zeros}}^{\mathrm{soft}} = \sum_i \exp\left[ - \left(\frac{\lambda_i}{\sigma_0 + \epsilon}\right)^2 \right]$.

A soft mask onto the positive spectrum assigns larger weights to eigenvalues farther from zero and normalizes them, that is, $w_i = 1 - \exp\left[ - \left(\frac{\lambda_i}{\sigma_0 + \epsilon}\right)^2 \right]$ and $\tilde{w}_i = \dfrac{w_i}{\sum_j w_j + \epsilon}$.

The mean on the positive spectrum is the corresponding weighted average, $\lambda_{\mathrm{mean},+}^{\mathrm{soft}} = \sum_i \tilde{w}_i \lambda_i$.

A weighted soft minimum over the positive spectrum uses a negative log-sum-exp with temperature $\tau > 0$, that is, $\lambda_{\mathrm{min},+}^{\mathrm{soft}} = -\tau \log \sum_i \exp(-\lambda_i/\tau) \tilde{w}_i$.

Note that $\varepsilon > 0$ is a small numerical constant for stability. The parameters $\epsilon, \sigma_0$ control the zero-tolerance window; $\tau$ tunes the softness of softmax/softmin (larger $\tau$ indicates closer to hard extremum). For numerical stability and differentiability, we applied a Gaussian gate around $\lambda = 0$, which provides a smooth surrogate for the connected components and yields soft weights $w_i$ for the positive spectrum, with denominators clamped by $\epsilon$. And we replace hard $\max / \min$ with log-sum-exp softmax/softmin (temperature $\tau$), optionally weighted by the positive-spectrum mask.

The standard deviation on the positive spectrum is computed from moments, but with a Huberized square-root to avoid gradient blow-ups at zero variance, which equals $0$ at $x = 0$ and has bounded gradient $1/(2\delta)$, that is, $\mathrm{var}_+ = \sum_i \tilde{w}_i \lambda_i^2 - \mu_+^2, \delta = \rho s, s = \mathrm{mean}(|\lambda|)$ (no grad), $\lambda_{\mathrm{std},+}^{\mathrm{soft}} = \sqrt{\mathrm{var}_+ + \delta^2} - \delta$, where $\rho$ scales the smoothing radius $\delta$ to the spectrum magnitude.

These safety techniques remove non-smooth operations, keep gradients well-behaved at zero/near-zero eigenvalues, and make the topo–spectral summary fully differentiable and efficient enough for end-to-end training, addressing a central challenge in topological deep learning.

Finally, we output

$$\mathbf{f}(L) = \left[ \lambda_{\mathrm{zeros}}^{\mathrm{soft}}, \ \lambda_{\max}^{\mathrm{soft}}, \ \lambda_{\mathrm{sum}}^{\mathrm{soft}}, \ \lambda_{\mathrm{mean},+}^{\mathrm{soft}}, \ \lambda_{\mathrm{std},+}^{\mathrm{soft}}, \ \lambda_{\mathrm{min},+}^{\mathrm{soft}} \right] \in \mathbb{R}^6.$$

In practice, we compute $\mathbf{f}$ per filtration scale and average (or learnable-weight) across scales; channels for different physicochemical roles are processed in parallel and concatenated.

### 3.2 MULTI-CHANNEL ENCODING OF PHYSICOCHEMICAL INTERACTIONS

To capture heterogeneous interaction modes at PPIs without hand-crafted heuristics, we encode each interface as a multi-channel graph built from role-aware atom types. Concretely, for each chain we map Atom37 names to eleven physicochemical-role–aware classes $G_1, \ldots, G_{11}$ (see Table.6 for illustrations), separating backbone donors/acceptors, aliphatic vs. aromatic carbons, basic nitrogens, carboxylate oxygens, hydroxyl oxygens, and sulfur atoms. This taxonomy isolates physicochemical roles that dominate PPIs (hydrogen bonding, salt bridges, hydrophobic packing, $\pi$–$\pi$, cation–$\pi$, and S-mediated contacts), preventing signal cancellation that often occurs in single-channel encodings.

Given interacting proteins $A$ and $B$, we construct cross-protein channel pairs $\mathcal{P} = \{(i,j) : i \in G^A, j \in G^B\}$ (default: full Cartesian product). This multi-channel view (i) factorizes interaction types at the graph level to yield cleaner spectral signatures, (ii) preserves interpretability by aligning

salient channels with biophysical modes, and (iii) in design settings leverages the Atom37 coupling between atom names/geometry and residue identity, so gradients on Atom37-level features can inform sequence updates.

In practice, we use attention to capture salient cross-channel interactions and learnable channel weights to select/amplify task-relevant types. When only Atom37 names are available, a physico-chemical–role–aware partition offers simple rules and clear semantics, serving as a standard preprocessing step for PPI interface representation and spectral–topological learning.

### 3.3 Binding Affinity Prediction Model

**Model Architecture**  TopoScorer performs binding affinity prediction from multi-scale topological spectral features of the protein interface, denoted $x$. Let $x \in \mathbb{R}^{B \times M \times S \times C}$, where $M$ denotes the per-scale statistic types, $S$ is the number of thresholded scales in ascending order, and $C$ is the physicochemical channels. The model treats the multi-scale features as a length-$S$ sequence. It first applies SENets(Li et al., 2017) over the $C$ channels to improves channel interdependencies, highlighting those most informative for the current sample. It then adopts a hybrid representation combining a low-rank bilinear projection with a residual full projection. The main branch maps $C \rightarrow d_{\mathrm{mid}}$ and then aggregates along $M$ to produce a $d_{\mathrm{model}}$-dimensional token embedding, while the auxiliary branch flattens $M \times C$ and maps directly to $d_{\mathrm{model}}$; the two are combined through a learnable small gate, so training begins along the low-rank path and gradually unlocks the residual capacity. The resulting scale tokens are equipped with learnable positional encodings and fed into a stack of Transformer encoder blocks (multi-head self-attention and feed-forward layers) to model cross-scale dependencies. After encoding, a LayerNorm and pooling (CLS pooling or mean pooling) are applied, followed by a two-layer MLP to output a scalar affinity score. The structural diagram of the model is shown in Fig.1. Detailed illustrations of modules in Appendix A.6.

### 3.4 Finetuning Andibody Design Model with TopoScorer

We finetune an antibody design model IgGM(Wang et al., 2025) on antibody–antigen complexes curated from SAbDab(Dunbar et al., 2014). Given an antigen sequence and an antibody sequence (with possibly incomplete CDRs), the base model jointly generates sequence and 3D structure via a diffusion process (Fig. 1). For each training example with ground-truth coordinates $X$, we sample a timestep $t$ and corrupt the antibody coordinates to obtain $X_t$, and optionally mask CDRs in the sequence. The model then denoises $(X_t, \mathrm{masked\ seq})$ to produce a prediction $\hat{X}$ and an updated antibody sequence; full-atom coordinates are recovered by side-chain packing from $\hat{X}$. We delineate the protein–protein interface on $\hat{X}$ and compute multi-channel, multi-scale topological–spectral features as in Secs. 3.1 and 3.2. A frozen TopoScorer maps these features to an affinity score that serves as a training reward. The overall objective combines standard structure/sequence losses with this affinity reward, encouraging geometrically accurate and higher-affinity designs. Detailed settings and a step-by-step training routine are provided in the Appendix A.5.

## 4 Experiments

### 4.1 Predicting Binding Affinity of PPI

We evaluated our affinity predictor on the PPB-Affinity test set using the score $-\log_{10} K_D$ (larger is better; Fig. 2(h)) and compared it with representative baselines: physics/energy–function methods (PyRosetta (Chaudhury et al., 2010) and FoldX Delgado et al. (2019)), a sequence-based model (ESM-1v (Meier et al., 2021a)), an unsupervised, SE(3)-equivariant generative energy model (DSM-Bind (Jin et al., 2023)), a structure-conditioned inverse-folding model (ESM-IF (Hsu et al., 2022b)), and a 3D structure–based interface predictor (PRODIGY (Xue et al., 2016)). To avoid interface-similarity–induced data leakage, we follow the recommendation of Bushuiev et al. and use the PPIRef tool to identify near-duplicate interfaces, defined as test interfaces that have at least one training interface with iDist distance less than 0.04; we then remove all such potential leakage cases from the test set. As summarized in Table 1, our method attains state-of-the-art Spearman and Pearson across all models on the leakage-filtered test set.

Table 1: Affinity, single-mutation, and multi-mutation performance (Spearman/Pearson). A dash indicates an unavailable metric. **Bold** indicates best performance, underline indicates second best.

| Method | Affinity Prediction | | Single-mutation | | Multi-mutations | |
|---|---|---|---|---|---|---|
| | Spearman | Pearson | Spearman | Pearson | Spearman | Pearson |
| PyRosetta | 0.1856 | 0.1954 | 0.3422 | 0.3285 | 0.2927 | 0.2258 |
| FoldX | 0.3295 | 0.3008 | 0.4355 | 0.4586 | 0.3734 | 0.3241 |
| ESM-1v | 0.1034 | 0.0876 | 0.1524 | 0.1921 | 0.1512 | 0.1736 |
| ESM-IF | 0.0530 | 0.0244 | 0.1116 | 0.1047 | 0.1697 | 0.0700 |
| PRODIGY | 0.1549 | 0.1277 | 0.3233 | 0.2902 | 0.3421 | 0.3236 |
| DSMBind | 0.3072 | 0.3269 | 0.3530 | 0.3261 | 0.3673 | 0.2954 |
| DDGPred | — | — | 0.5522 | 0.5303 | 0.4585 | 0.5638 |
| RDE-Network | — | — | 0.5127 | **0.6067** | 0.5397 | **0.6108** |
| GearBind | — | — | 0.5014 | 0.5496 | 0.5470 | 0.5616 |
| TopoNetTree | — | — | 0.5185 | 0.5508 | — | — |
| TopoScorer | **0.3848** | **0.3804** | **0.5876** | 0.5615 | **0.5704** | 0.5652 |

To further validate performance, we trained our model on SKEMPI 2.0 and evaluated on test set curated from the single-mutation subset and the multi-mutation subset of SKEMPI 2.0, with the same procedure of preventing data leakage as decribed above. All test-set complexes are strictly disjoint from the training data: no complex appearing in the test set is present in the training set. In addition to the above baselines, we included the current state-of-the-art end-to-end predictors DDG-Pred(Shan et al., 2022), the pretrained flow model RDE-Network(Luo et al., 2023) , all atom based graph model GearBind(Cai et al., 2024) and a previous persistent homology based model TopoNet-Tree(Wang et al., 2020). As shown in Table1, Our approach reached state-of-the-art Spearman correlations and second best Pearson on single-mutation and multi-mutation subsets. The strong Spearman indicates our superior ranking ability. Prior evaluations of Boltz-2(Passaro et al., 2025) show that affinity prediction performance varies markedly across assays and can be confounded by errors in predicted structures, limited generalization to unseen protein families, and sensitivity to out-of-distribution small molecules; under such variability, ranking accuracy (Spearman) is typically the more robust indicator of practical utility. Notably, The competing RDE-Network relies on large pretrained components (about $133M + 63M$ parameters), whereas our model has only $\sim 43M$ parameters while keeping interpretability. These results show that our lightweight model attains strong ranking performance and comparable prediction accuracy, indicating that the proposed topological feature extraction effectively captures information relevant to binding affinity. We measured the inference time per sample on the single-mutation task, using the same CPU(Fig 2(k)). TopoScorer achieves fast and stable inference time($5.01 \pm 0.1ms$ per sample) among compared models. DSM-Bind is also computationally efficient, but its predictive accuracy on our benchmarks is substantially lower than TopoScorer, so it does not offer the same balance of speed and reliability.Taken together, TopoScorer strikes a rare and favorable balance between predictive accuracy, parameter efficiency, and inference speed, which is not achieved by the other methods we compare against. More details in AppendixA.7 and A.6.

## 4.2 TopoScorer-Guided Fine-Tuning Improves Antibody Sequence–Structure Co-Design

To assess the impact of introducing TopoScorer as a differentiable interface affinity signal, we used IgGM(Wang et al., 2025), which is a antibody sequences and structures co-design model, as the base model and constructed a held-out test set of 763 protein–protein complexes from SAbDab released after December 30, 2023 (thereby ensuring no overlap with the data used to train the base model). We then compared the TopoScorer-fine-tuned model against the baseline IgGM with the authors' publicly released weights. The results are summarized in Table 2.

In the structure-only setting, the fine-tuned model delivers a clear lift in interface quality: DockQ increases by 20.8% and $SR(DockQ > 0.23)$ rises by 31.24%. In sequence–structure co-design, DockQ again trends upward across splits (on the order of $\sim 20\%$ on average), with an average reduction of approximately 4% outside the H2 split and a larger decrease when all CDRs are masked. In general, the addition of TopoScorer steers the optimization towards more plausible interface

Table 2: Sequence and structure co-design of antibodies for specific antigen. Arrows indicate directionality: higher is better (↑), lower is better (↓). **SR**=% with DockQ>0.23; **Bold** indicates better performance.

| Model | CDR | AAR↑ | RMSD(Cα)↓ | DockQ↑ | SR(%)↑ |
|-------|-----|------|-----------|--------|--------|
| w/ Topo | Struct. | / | **2.1807** | **0.1754**(+20.88%) | **27.26**(+31.24%) |
| | H1 | 0.3934 | **2.2876** | **0.1492**(+101.9%) | **21.90**(+5.44%) |
| | H2 | **0.3925** | **2.2902** | **0.1439**(+1.91%) | **20.17**(+1.15%) |
| | H3 | **0.3833** | **2.3153** | **0.1232**(+0.735%) | 14.08(-1.745%) |
| | L1 | 0.3561 | **2.4109** | **0.0992** (+3.766%) | **12.78**(+6.589%) |
| | L2 | **0.3568** | **2.4102** | **0.1078**(+19.12%) | 12.85(-35.56%) |
| | L3 | **0.3564** | **2.3851** | **0.1000**(+2.56%) | 13.43(-1.395%) |
| | All | **0.3408** | **2.3943** | **0.0787**(6.495%) | **8.30**(+10.08%) |
| Base | Struct. | / | 2.3371 | 0.1451 | 20.77 |
| | H1 | **0.5320** | 2.3026 | 0.0739 | 20.77 |
| | H2 | 0.3916 | 2.2959 | 0.1412 | 19.94 |
| | H3 | 0.0469 | 2.3284 | 0.1223 | **14.33** |
| | L1 | **0.7329** | 2.4469 | 0.0956 | 11.99 |
| | L2 | 0.3343 | 2.4313 | 0.0905 | **19.94** |
| | L3 | 0.2844 | 2.4237 | 0.0975 | **13.62** |
| | All | 0.3256 | 2.4384 | 0.0739 | 7.54 |

geometry while markedly improving recoverability at the most challenging site (H3); when the topology–geometry objective conflicts with native-sequence matching, AAR may drop (e.g., H1, L1), reflecting the trade-off of "better interface" vs. "closer-to-native sequence."

### 4.3 INTERPRETABILITY ANALYSIS

To probe how spatial scale, channel, and spectral statistics shape affinity prediction, we computed heatmaps of output gradients along these axes. In the scale axis (Fig. 2(a)), attribution peaks at ∼3.5–4.0 Å, coinciding with the first percolation of the interface contact graph where spectral summaries are most perturbed. Across spectral descriptors (Fig. 2(b)), the sum of eigenvalues dominates, implicating total interfacial connectivity as the primary driver. Cross-channel maps (Fig. 2(c)) show side-chain features consistently outweighing backbone terms; Lys/Arg cationic nitrogens act as hubs with broad coupling, highlighting side-chain–mediated cation anchoring as a key determinant.

Focusing on the most salient feature—the maximum eigenvalue, we analyzed and compared the resulting topological spectral signatures for the 1ACB wild type and its two point mutants(Fig.2(d)). Substituting L38 from leucine (L) to aspartate (D) or glutamate (E) replaces a hydrophobic side chain with a carboxylate, disrupting packing and adding acceptor oxygens. For cross-protein carboxylate-oxygen neighbors (Fig.2(e)), the wildtype exhibits a later onset and lower step-like PTHL/Laplacian curve, indicating a sparser, less connected like-charge subgraph. The aspartate/glutamate mutants rise earlier and higher, consistent with increased repulsive carboxylate oxygens contacts; accordingly, the spectral sum grow and connected components merge sooner, matching the affinity drop. The mutations also introduce an anionic partner for receptor $N^+$ groups, so carboxylate oxygen–cationic nitrogen neighbors appear at smaller thresholds and increase rapidly(Fig.2(f)). Glutamate has one extra methylene relative to aspartate, giving greater reach and flexibility to satisfy favorable carboxylate oxygen–cationic nitrogen geometry; thus the glutamate mutant is highest, the aspartate mutant lower, and the wild type lowest (the wild type lacks this anionic site). Nevertheless, the net affinity still decreases, because pocket disruption and geometric penalties outweigh the salt-bridge gains. In the wild type, leucine at position 38 forms a hydrophobic pocket that seats and orients lysine/arginine side chains, yielding more cationic nitrogen-aliphatic carbon neighbors and a higher curve(Fig.2(g)). Converting leucine to aspartate or glutamate weakens this pocket; but glutamate's extra methylene acts as a hydrophobic spacer that extends the negative charge outward while retaining nearby aliphatic contacts, making the glutamate curve closer to the wild type in the cationic nitrogen-aliphatic carbon motif. More results shown in Appendix Fig.3 and 4. We provide additional case studies in Appendix A.8.3.

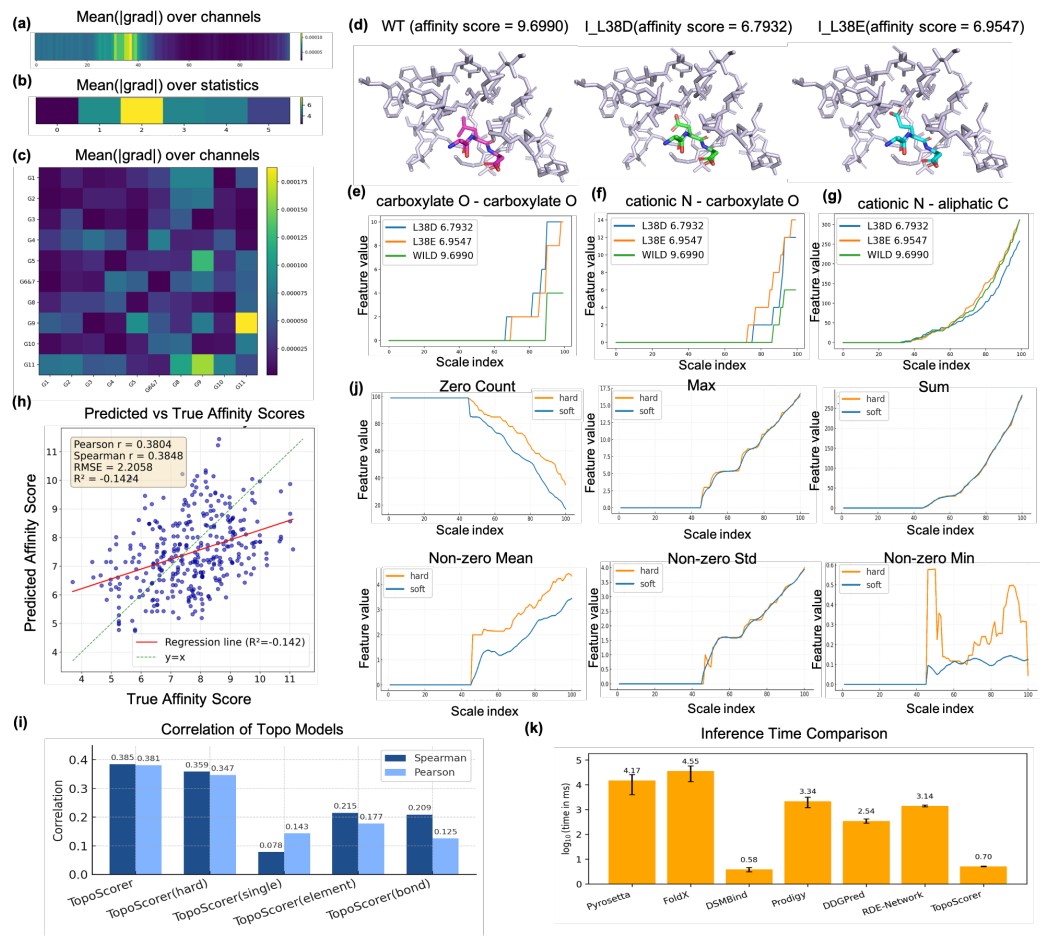

Figure 2: (a) Heatmap of filters. (b) Heatmap of spectral features. (c) Heatmp of physicochemical channels. (d) Structures of 1ACB and mutants residues 37–39 (pink: wild type ligand chain I; green: mutant L38D ligand chain I; blue: mutant L38E ligand chain I; light purple: receptor chain E). (e) $\lambda_{sum}$ between receptor carboxylate oxygens and ligand carboxylate oxygens. (f) $\lambda_{sum}$ between receptor cationic nitrogens and ligand carboxylate oxygens. (g) $\lambda_{sum}$ between receptor cationic nitrogens and ligand aliphatic carbons . (h) Performance of TopoScorer on PPB-Affinity. (i) Results of Ablation Studies. (j) Comparison of topo-spectral features obtained by soft and hard extraction. (k) Results of inference time comparison(in log scale).

## 4.4 ABLATION STUDIES

### 4.4.1 EFFECTS OF PHYSICOCHEMICAL MULTI-CHANNEL ENCODING

To assess the effectiveness of our physicochemical cross-channel encoding in TopoScorer, we compared it with two baselines: (i) a single-channel variant that computes topological spectra from all PPI atoms in one channel; and (ii) an element-wise variant that partitions PPI atoms into $\{(C), (N), (O), (S), (C, N), (C, O), (C, S), (N, O), (N, S), (O, S), (C, N, O, S)\}$ and computes topological spectra features for all pairwise combinations; and (iii) an bond-count variant that counts the number of interfacial contacts ("bonds") between the corresponding atom groups. As shown in Fig.2(i), under identical training/testing protocols, our cross-channel encoding attains Spearman's $\rho = 0.3848$ and Pearson's $r = 0.3804$, outperforming the single-channel model ($0.0783/0.1431$), the element-wise model ($0.2147/0.1773$) and the bond-count model($0.209/0.1252$). We attribute these gains to the explicit modeling of inter-channel physicochemical interactions—e.g., hydrophobic–polar, donor–acceptor, and cation–$\pi$ contacts—and the ability to adaptively reweight informative channels; in contrast, the single-channel baseline collapses interaction structure, and the

element-wise grouping—though stronger—ignores residue-level roles and thus cannot fully capture functional interactions.

### 4.4.2 Effects of Soft-Thresholding and Approximate Statistical Analysis

We compare the spectral features produced by our differentiable topological spectral feature extraction with those obtained using existing non-differentiable methods(eg. used in (Chen et al., 2024)) (see Fig. 2(j)). The traditional pipeline computes spectra by exact counting of eigenvalue–based statistics, whereas our differentiable formulation can be regarded as a smooth approximation to it. Empirically, the curves for the maximum, sum, and variance of non-zero entries closely track the traditional results. The zero count and the mean of non-zero entries show offsets in magnitude but capture the same overall trends. The minimum of non-zero entries exhibits the largest discrepancy: the discrete method fluctuates markedly, while our differentiable features change more smoothly.

We additionally trained an affinity prediction model using topological spectral features obtained with the hard-threshold (non-differentiable) pipeline to evaluate the impact of the approximation introduced by our differentiable spectral-statistics analysis. As shown in Fig. 2(i), the differentiable approximation yields results comparable to the traditional exact method; any potential error does not adversely affect affinity prediction, likely because the error distributions in the training and test sets are similar and largely cancel out, and the non-zero-mean features with the largest errors constitute only a small fraction of the predictions(Fig 2).

## 5 Conclusions

In this work, we introduced TopoScorer, a lightweight and interpretable affinity scorer that is fully differentiable and can be used as a loss or reward to steer generative protein models. It is comprimised of Specter, a differentiable, multi-channel, multi-scale topo-spectral features extractor for protein–protein interfaces (PPIs). Across two public benchmarks, TopoScorer delivers performance comparable to state-of-the-art methods, and ablations highlight the contribution of our topo-spectral features. When integrated to finetune a state-of-the-art antibody design model, TopoScorer improves metrics of the generated candidates. We further provide interpretability analyses that link the learned spectral statistics to physicochemical properties of interfaces.

### Ethics Statement

This work uses only publicly available protein-structure and binding-affinity datasets and code; no human-subject data or animal experiments were conducted, and no personally identifiable information is involved.

### Reprodicibility Statement

Source codes and data are provided in `https://anonymous.4open.science/r/Anonymous_code-DD4E`.

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

# A APPENDIX

## A.1 USAGE OF LLMS

In this work, we used **ChatGPT-5** (OpenAI) to assist with writing guidance and language polishing.

## A.2 TOPOLOGICAL BASICS

**Graphs.** A (simple) graph $G = (V, E)$ consists of a finite vertex set $V$ and an edge set $E \subseteq \{\{u, v\} : u, v \in V, u \neq v\}$; edges connect unordered pairs of distinct vertices. Variants include *directed* graphs, where $E \subseteq V \times V$ consists of ordered pairs, and *weighted* graphs, where each edge $e$ carries a weight $w_e > 0$. With a fixed ordering of vertices, the adjacency matrix $A \in \mathbb{R}^{|V| \times |V|}$ has entries $A_{uv} = w_{uv}$ (or 1 in the unweighted case) when $\{u, v\} \in E$; the degree matrix is $D = \mathrm{diag}(d(v))$ with $d(v) = \sum_u A_{vu}$. The (combinatorial) graph Laplacian is $L = D - A$, a symmetric positive semidefinite operator whose kernel encodes connected components.

**Hypergraphs.** A hypergraph $\mathcal{H} = (V, \mathcal{E})$ generalizes a graph by allowing each hyperedge $e \in \mathcal{E}$ to be an arbitrary nonempty subset $e \subseteq V$ of any cardinality. When all hyperedges share the same size $k+1$, $\mathcal{H}$ is called $(k+1)$-uniform. Weighted hypergraphs assign a positive weight $w_e$ to each $e$, and directed hypergraphs may specify ordered hyperedges (e.g., $e = (S \to T)$ with disjoint source/target vertex sets). A convenient linear-algebraic representation uses the vertex–hyperedge incidence matrix $H \in \{0, 1\}^{|V| \times |\mathcal{E}|}$ with $H_{ve} = 1$ iff $v \in e$, together with diagonal degree matrices $D_V = \mathrm{diag}(d(v))$ and $D_E = \mathrm{diag}(|e|)$, and an optional weight matrix $W = \mathrm{diag}(w_e)$. These ingredients yield normalized diffusion operators such as $L_{\mathrm{sym}} = I - D_V^{-1/2} H W D_E^{-1} H^\top D_V^{-1/2}$, which reduce to the graph Laplacian when every hyperedge has size two. In applications, hypergraphs model higher-order (multiway) relations that cannot be expressed as pairwise edges; common reductions to graphs include the clique (flag) expansion, which replaces each hyperedge by a clique on its vertices, and the star expansion, which forms a bipartite graph between $V$ and $\mathcal{E}$ via $H$.

**Simplicial Complex** A simplicial complex provides a parsimonious, algebraically tractable representation of geometric structure by assembling "simple pieces" (simplices) into a topological space. Formally, a $k$-simplex is the convex hull of $k+1$ affinely independent vertices (abstractly, a $(k+1)$-subset of a vertex set), with faces given by all vertex-subsets; an (abstract) simplicial complex $K$ on vertex set $V$ is a family of finite subsets of $V$ that is closed under taking subsets, and its dimension is $\dim K = \max_{\sigma \in K}(|\sigma| - 1)$. Equipped with orientations, simplices generate chain groups $C_k$ with boundary operators $\partial_k : C_k \to C_{k-1}$, yielding homology groups $H_k$ that quantify $k$-dimensional holes (components, cycles, voids). In practice, complexes are built from data via metric or combinatorial rules and then organized into a filtration $\emptyset = K_{t_0} \subseteq K_{t_1} \subseteq \cdots \subseteq K_{t_m}$ over a scale parameter $t$, enabling multiscale analysis and persistent homology.

**Constructing Simplicial Complexes** Simplicial complexes provide combinatorial surrogates of metric or geometric data and are typically organized into filtrations to support multiscale analysis via persistent homology. Among the standard constructions, the Vietoris–Rips (VR) complex offers the most accessible entry point: at scale $r > 0$, one builds a graph on the data with edges between points at distance $\leq r$ and then takes its flag (clique) completion; equivalently, a $k$-simplex belongs to $\mathrm{VR}(X; r)$ precisely when all pairwise distances among its vertices are at most $r$. Because it depends only on pairwise distances, VR does not require coordinates and is straightforward to implement from $k$-NN or radius graphs. Its chief limitation is combinatorial growth in high dimensions, which practitioners mitigate by truncating to low homological degrees (e.g., $k = 0, 1, 2$) and by relying on the filtration $\{ \mathrm{VR}(X; r) \}_{r \geq 0}$ rather than any single threshold.

A more geometrically faithful alternative is the Čech complex. Fixing radius $r/2$, one places closed balls around each point and inserts a simplex whenever the corresponding balls have a nonempty common intersection. The nerve theorem guarantees that $\check{\mathrm{Cech}}(X; r)$ is homotopy equivalent to the union of these balls, so the complex closely reflects the topology of the offset shape. In Euclidean settings, the Čech and VR filtrations are interleaved (with constants depending on the ambient metric), which justifies using the computationally cheaper VR filtration while retaining stability guarantees for persistence. When coordinates are available and geometric sharpness matters, Alpha

complexes refine this picture through Delaunay geometry: a simplex enters the complex when its circumscribed empty ball (or, in the weighted case, power ball) has radius at most $r/2$. The resulting Alpha filtration tends to be smaller and more parsimonious than VR at comparable scales and is particularly effective for shapes with meaningful cavities and tunnels, such as molecular surfaces, where weighted variants incorporate atom-specific radii.

For very large data sets, Witness and Lazy Witness complexes trade exactness for scalability by introducing a small landmark set $L \subset X$ and using the remaining points as witnesses to certify simplices on $L$. A simplex is admitted when some witness is sufficiently close to all of its vertices under a chosen proximity rule; the lazy variant first constructs witnessed edges and then takes the flag completion for higher-dimensional simplices. This sparsification preserves the global topological signal with far fewer vertices, provided landmarks are selected judiciously (e.g., max–min sampling or $k$-means centers) and proximity parameters are tuned to data scale.

Across these constructions, the common practice is to form a filtration $\{K_r\}_{r \geq 0}$ by increasing the scale parameter and to summarize multiscale topology using barcodes or persistence diagrams. In applications, VR is the default choice when only distances are available or when ease of implementation is paramount; Čech is preferred when recovering the topology of offsets is critical; Alpha excels when coordinate accuracy and geometric features (voids, tunnels) drive the analysis, especially with weighted variants; and Witness complexes enable exploratory or large-scale TDA under tight memory and time budgets, all while remaining compatible with stable persistent homology pipelines.

### A.3 TOPOLOGICAL HYPERGRAPHS AND PTHLS

**From hypergraphs to topology.** While incidence-based Laplacians encode diffusion on $V$, they do not, by themselves, expose higher-order "holes" (cycles, voids) created by multi-way relations. A *topological hypergraph* endows $\mathcal{H}$ with an algebraic-topological structure by organizing hyperedges into graded families and equipping them with orientations so that boundary and coboundary operators can be defined. Concretely, fix an integer $k_{\max} \geq 1$ and define $k$-cells as oriented $(k{+}1)$-element hyperedges. Write $C_k$ for the real vector space spanned by all oriented $k$-cells, and assemble linear boundary maps

$$\partial_k : C_k \longrightarrow C_{k-1}, \qquad \partial_k(v_0, \ldots, v_k) = \sum_{i=0}^{k} (-1)^i [v_0, \ldots, \widehat{v_i}, \ldots, v_k],$$

whenever all faces on the right-hand side are present as $(k{-}1)$-cells. This yields a chain complex

$$\cdots \xrightarrow{\partial_{k+1}} C_k \xrightarrow{\partial_k} C_{k-1} \xrightarrow{\partial_{k-1}} \cdots \xrightarrow{\partial_1} C_0, \qquad \partial_k \circ \partial_{k+1} = 0,$$

and therefore homology groups $H_k$ that quantify $k$-dimensional voids generated by multi-way interactions. In matrix form, take $B_k$ to be the signed incidence matrix of $\partial_k$ after fixing an ordering of $k$- and $(k{-}1)$-cells. With positive diagonal weight matrices $W_k$ on $k$-cells one obtains the *Hodge hypergraph Laplacians*

$$L_k = B_k^\top W_{k-1} B_k + B_{k+1} W_{k+1} B_{k+1}^\top,$$

whose kernel $L_k$ is isomorphic to the $k$-th homology (the space of harmonic $k$-forms), while nonzero spectra capture "gradient" and "curl" energies of $k$-signals. This construction mirrors the simplicial Hodge theory but keeps the modeling focus on hyperedges rather than requiring a full clique completion.

**From boundary/coboundary to the Hodge Laplacian.** Let $K$ be a finite oriented simplicial complex. Fix an ordering of $k$-simplices and write $C_k(K; \mathbb{R})$ for the $k$-chain space with the usual boundary maps

$$\partial_k : C_k \longrightarrow C_{k-1}, \qquad \partial_k \circ \partial_{k+1} = 0.$$

In coordinates, $\partial_k$ is represented by the signed incidence matrix $B_k$ (rows index $(k{-}1)$-simplices, columns index $k$-simplices). The $k$-*cochain* space is $C^k = \mathrm{Hom}(C_k, \mathbb{R})$, and the *coboundary* $\partial_k : C^k \to C^{k+1}$ is the algebraic adjoint of $\partial_{k+1}$, hence in the standard bases

$$\partial_k = B_{k+1}^\top.$$

To turn adjoints into matrices, equip each $C^k$ with an inner product $\langle \cdot, \cdot \rangle_k$ induced by a symmetric positive definite matrix $W_k$ (often diagonal, encoding $k$-cell weights): for $\alpha, \beta \in C^k \simeq \mathbb{R}^{n_k}$,

$$\langle \alpha, \beta \rangle_k = \alpha^\top W_k \beta.$$

The adjoint $\partial_k^* : C^{k+1} \to C^k$ is defined by $\langle \partial_k \alpha, \beta \rangle_{k+1} = \langle \alpha, \partial_k^* \beta \rangle_k$ for all $\alpha, \beta$. In matrices this becomes

$$(B_{k+1}^\top \alpha)^\top W_{k+1} \beta = \alpha^\top W_k (\partial_k^* \beta) \quad \forall \alpha, \beta \implies \partial_k^* = W_k^{-1} B_{k+1} W_{k+1}.$$

Similarly, $\partial_{k-1} = B_k^\top$ and $\partial_{k-1}^* =, W_{k-1}^{-1} B_k W_k$. The (combinatorial) $k$-*Hodge Laplacian* acting on $k$-cochains is

$$\Delta_k = \partial_k \partial_k^* + \partial_{k+1}^* \partial_{k+1}.$$

Substituting the matrices and simplifying yields the explicit weighted form

$$\Delta_k = B_k^\top W_{k-1}^{-1} B_k W_k + W_k^{-1} B_{k+1} W_{k+1} B_{k+1}^\top \qquad \text{(self-adjoint w.r.t. } \langle \cdot, \cdot \rangle_k \text{).}$$

When one prefers a symmetric matrix under the *Euclidean* inner product, it is convenient to conjugate by $W_k^{1/2}$ and work with

$$L_k = W_k^{1/2} \Delta_k W_k^{-1/2} = \underbrace{\left( W_k^{1/2} B_k^\top W_{k-1}^{-1} B_k W_k^{1/2} \right)}_{\text{"lower" part } (\partial_{k-1} \partial_{k-1}^*)} + \underbrace{\left( W_k^{-1/2} B_{k+1} W_{k+1} B_{k+1}^\top W_k^{-1/2} \right)}_{\text{"upper" part } (\partial_k^* \partial_k)},$$

which is symmetric positive semidefinite in the usual sense and unitarily similar to $\Delta_k$.

**Unweighted special case.** If $W_k = I$ for all $k$ (orthonormal bases of cochains), then $\partial_k^* = B_{k+1}$ and

$$\Delta_k = B_k^\top B_k + B_{k+1} B_{k+1}^\top \qquad \text{(standard combinatorial Hodge Laplacian).}$$

In either weighted or unweighted form, the Hodge decomposition follows: $\ker \Delta_k \cong H^k(K; \mathbb{R})$ (harmonic $k$-cochains represent cohomology), while the ranges of $\partial_{k-1}$ and $\partial_k^*$ are orthogonal and encode "gradient" and "curl" subspaces of $k$-signals. These identities arise directly from $\partial_k \partial_{k+1} = 0$ and the definitions of adjoints with respect to the chosen inner products.

**Geometric filtrations and persistence.** To probe topology across scales, equip each hyperedge with a filtration value via a data-driven rule, for instance

$$w_e(r) = \prod_{(p,q) \in \text{pairs}_\times(e)} \kappa_\tau\big(d(p,q); r\big), \qquad \kappa_\tau(d; r) = \sigma\left( \frac{r-d}{\tau} \right),$$

where $d(\cdot, \cdot)$ is a metric on embedded vertices, $\sigma$ is a smooth step (e.g., logistic), $r$ is a scale, and $\tau > 0$ controls softness. Increasing $r$ generates a nested family of topological hypergraphs with boundary matrices $B_k$ and Hodge Laplacians $L_k(r)$. The matrix $B_k$ is the signed incidence (boundary) matrix of the oriented simplicial complex (or the precomputed "super–complex"). First enumerate all candidate $k$-simplices $\sigma = [v_0, \ldots, v_k]$ with a fixed orientation (e.g., vertices in ascending index order), and all $(k-1)$-simplices $\tau$. Its entries record face relations with alternating signs:

$$(B_k)_{\tau, \sigma} = \begin{cases} (-1)^i, & \text{if } \tau = \sigma \setminus \{v_i\} \text{ for some } i \in \{0, \ldots, k\}, \\ 0, & \text{otherwise.} \end{cases}$$

In the soft–boundary setup, this $B_k$ is constructed once from a radius upper bound (or the union of complexes over multiple radii) and kept fixed; the scale dependence is carried by diagonal membership/weight matrices rather than by changing $B_k$ itself.

One can compute persistent homology on the induced chain complex or summarize spectra $\{\lambda_i(L_k(r))\}_r$, obtaining stable multiscale descriptors. The soft kernel $\kappa_\tau$ makes $B_k(r)$ and $L_k(r)$ differentiable in $r$ and in the underlying coordinates, enabling end-to-end learning with topological regularizers or losses.

**Relations to clique/star expansions.** Common graph reductions of a hypergraph—clique (flag) expansion and star expansion—map every hyperedge to a clique on $V$ or to a bipartite star between $V$ and $\mathcal{E}$, after which one applies graph or simplicial homology. Although convenient, these expansions can introduce dense spurious high-order simplices and blur the combinatorics of the original multi-way relation. Topological hypergraphs, by contrast, keep $k$-cells exactly where $k{+}1$-wise interactions are modeled, yielding leaner chain groups and more faithful $B_k$ matrices, which often improves interpretability and computational efficiency for $k \geq 2$.

**Directed and weighted variants.** Many applications require orientation beyond sign conventions. A *directed* hypergraph allows ordered hyperedges $e = (S \to T)$ with disjoint source/target vertex sets. One extends the chain complex by declaring 1-cells to be ordered pairs and higher cells to be ordered tuples, then defining $\partial_k$ by alternating sums that respect direction. Weights can encode frequency, confidence, or physical strength; incorporating them in $W_k$ preserves the Hodge decomposition and yields anisotropic diffusion on $k$-signals. When vertices carry coordinates or attributes, mixed weights $W_k(\theta)$ can be learned jointly with downstream objectives.

## A.4 Geometric meanings of PTHLs spectra statistics

**Zero count of eigen values** Zero count of eigen values approximates the dimension of the harmonic subspace $\ker L_k^{(i,j)}(r_t)$, i.e., the $k$-order "holes/cycles" (Betti number surrogate) in the directed hypergraph at scale $r_t$. For $k = 0$ this corresponds to the number of cross-protein connected components; for $k \geq 1$ it reflects higher-order cycle-like interaction patterns among $(k{+}1)$-tuples across the interface.

**Smallest positive eigenvalue** Smallest positive eigenvalue measures the spectral gap above the harmonic space, i.e., the "cohomological connectivity" of $k$-order structures. Larger gaps imply more robust $k$-order coupling and fewer near-harmonic defects.

**Sum, mean and Variance** Sum, mean and Variance of non-zero eigen values summarize the overall "oscillation energy" and its spread at order $k$. For $k = 0$, sum of eigenvalues equals the total (weighted) degree and tracks aggregate interfacial proximity. For $k \geq 1$, the moments encode how strongly $k$-faces are bounded by $(k{-}1)$-faces and how they bound $(k{+}1)$-faces (through $\partial_k$ and $\partial_{k+1}$), reflecting the stiffness and heterogeneity of higher-order organization.

**Maximum eigenvalue** Controls the worst-case curvature of $k$-order diffusions/regularizers on the hypergraph, bounding step sizes in gradient flows and indicating the strongest local constraints at order $k$.

## A.5 Finetuning procedure with TopoScorer

We finetune an antibody design model IgGMWang et al. (2025) on 13,013 antibody–antigen complexes curated from SAbDab(Dunbar et al., 2014). We use samples before 2023-06-30 for training, samples from 2023-06-30 to 2023-12-30 for validation, samples after 2023-12-30 for testing. To reduce redundancy, antibody sequences in the training set are clustered by CD-HIT(Li & Godzik, 2006) at 95% sequence identity, yielding 3,815 sequence clusters. The trainer samples uniformly across clusters. We finetune the base model for three epochs on 8×A100 GPUs. The entire finetuning procedure is summarized in Algorithm. 1.

We also apply sequence and structure losses as follows:

**Frame-Aligned Point Error (FAPE).** FAPE measures pointwise discrepancies after aligning both prediction and ground truth in each residue's local rigid frame, making it insensitive to global rigid motions:

$$\mathcal{L}_{\text{fape}} = \frac{1}{N} \sum_i \sum_{a \in \mathcal{N}(i)} \min\Big(\tau, \ \big\| F_i(\hat{x}_a) - F_i(x_a) \big\| \Big),$$

where $F_i$ transforms coordinates into residue $i$'s local frame, $\mathcal{N}(i)$ is a neighborhood (e.g., backbone/side-chain points), and $\tau$ is a truncation radius.

---

**Algorithm 1:** Finetuning procedure with TopoScorer

---

**Input:** Antigen sequence $s^{\mathrm{Ag}}$; antibody sequence $s^{\mathrm{Ab}}$ (CDR may be incomplete); GT antibody backbone $X_{\mathrm{bb}}^{\mathrm{Ab}} \in \mathbb{R}^{L \times 3}$; GT full-atom $X$; noise schedule $\{\alpha_t, \sigma_t\}$; CDR masker $M_k$; packer $\mathcal{P}$; model $f_\theta$; interface extractor $\mathcal{I}$; channel selectors $\{\Pi_c\}_{c=1}^C$; radii set $\mathcal{R}$; soft kernel $\kappa_\tau$

**Output:** Loss $\mathcal{L}$ for backpropagation

Sample $t \sim p(t)$, $\varepsilon \sim \mathcal{N}(0, I)$, $k \sim \mathrm{CDR\_choice}$

Backbone noising and CDR masking: $X_{\mathrm{bb},t}^{\mathrm{Ab}} \leftarrow \alpha_t X_{\mathrm{bb}}^{\mathrm{Ab}} + \sigma_t \varepsilon$, $\tilde{s}^{\mathrm{Ab}} \leftarrow M_k(s^{\mathrm{Ab}})$

Denoise: $(\hat{X}_{\mathrm{bb}}, \hat{s}^{\mathrm{Ab}}) \leftarrow f_\theta(s^{\mathrm{Ag}}, \tilde{s}^{\mathrm{Ab}}, X_{\mathrm{bb},t}^{\mathrm{Ab}}, t)$

Sidechain packing: $\hat{X} \leftarrow \mathcal{P}(\hat{X}_{\mathrm{bb}}, \hat{s}^{\mathrm{Ab}})$

Interface extraction: $\hat{X}_{\mathrm{PPI}} \leftarrow \mathcal{I}(\hat{X})$

**for** $c = 1 \ldots C$ **do**

    $A_c^{\mathrm{Ag}} \leftarrow \Pi_c(\hat{X}_{\mathrm{PPI}}^{\mathrm{Ag}})$;    $A_c^{\mathrm{Ab}} \leftarrow \Pi_c(\hat{X}_{\mathrm{PPI}}^{\mathrm{Ab}})$

    $D_c \leftarrow \mathrm{pairwise\_dist}(A_c^{\mathrm{Ag}}, A_c^{\mathrm{Ab}})$

    **for** $r \in \mathcal{R}$ **do**

        $W_c^{(r)} \leftarrow \kappa_\tau(D_c; r)$

        $L_c^{(r)} \leftarrow \mathrm{PTHL}(W_c^{(r)})$

        $\phi_c^{(r)} \leftarrow \mathrm{SpecStats}(L_{0,c}^{(r)})$

$\Phi \leftarrow \mathrm{concat}\big(\{\phi_c^{(r)}\}_{c,r}\big)$;    $A \leftarrow \mathrm{AffinityPred}(\Phi)$

$\mathcal{L} \leftarrow \mathcal{L}_{\mathrm{fape}}(X, \hat{X}) + \mathcal{L}_{\mathrm{mse}}(X, \hat{X}) + \mathcal{L}_{\mathrm{lddt}}(X, \hat{X}) + \mathcal{L}_{\mathrm{srcv}}(X, \hat{X}) + \mathcal{L}_{\mathrm{viol}}(\hat{X}) - A$

---

**Local Distance Difference Test (lDDT) loss.** lDDT assesses preservation of local pairwise distances within tolerance thresholds; we minimize $1 - \mathrm{lDDT}$:

$$\mathrm{LDDT}(\hat{x}, x) = \frac{1}{|\mathcal{P}|} \sum_{(i,j) \in \mathcal{P}} \frac{1}{4} \sum_{\delta \in \{0.5, 1, 2, 4\}\,\text{Å}} \mathbf{1}\Big( \big| \|\hat{x}_i - \hat{x}_j\| - \|x_i - x_j\| \big| < \delta \Big), \qquad \mathcal{L}_{\mathrm{lddt}} = 1 - \mathrm{LDDT}.$$

**Violation loss.** Similar to AlphaFold2 Jumper et al. (2021), we introduce penalty terms for (i) incorrect peptide *bond length* and *bond angles*, and (ii) steric *clashes* between non-bonded atoms. For multimer structure prediction, we do not penalize the bond length and angle between the last residue in the heavy chain and the first residue in the light chain, since there is no peptide bond between them. In addition, following AlphaFold-Multimer Evans et al. (2021a), we normalize the steric-clash penalty by the number of non-bonded atom pairs that are in clash to stabilize optimization. The overall loss is

$$\mathcal{L}_{viol} = \mathcal{L}_{bond\text{-}length} + \mathcal{L}_{bond\text{-}angle} + \mathcal{L}_{clash}. \tag{1}$$

Here, $\mathcal{L}_{bond\text{-}length}$ penalizes deviations of predicted peptide bond lengths from their canonical targets, $\mathcal{L}_{bond\text{-}angle}$ penalizes deviations of backbone bond angles, and $\mathcal{L}_{clash}$ penalizes steric overlaps between non-bonded atom pairs, with the clash term normalized by the number of clashing pairs.

**Amino-acid Sequence Recovery Loss ($\mathcal{L}_{\mathbf{srcv}}$).** To supervise the model to recover the amino-acid identity $s_i$ at each design/masked position $i$, we formulate a 20-way classification over the standard amino acids. The representation at position $i$ is linearly projected to class probabilities $\{p_i^c\}_{c=1}^{20}$, and the objective is the cross-entropy loss:

$$\mathcal{L}_{\mathrm{srcv}} = -\frac{1}{L_{\mathrm{design}}} \sum_{c=1}^{20} p_i^c \log y_i^c, \tag{2}$$

where $p_i^c$ denotes the predicted probability of class $c$ at position $i$, $y_i^c$ is the one-hot ground-truth label ($y_i^{c^\star} = 1$ for the true class $c^\star$ and 0 otherwise), and $L_{\mathrm{design}}$ is the set of design/masked positions over which the loss is averaged.

**Mean squared error (MSE).** For continuous supervision (e.g., affinity, energies, spectral statistics), we use MSE:

$$\mathcal{L}_{\mathrm{mse}} = \frac{1}{N} \sum_{n=1}^{L} \left( \hat{x}_n - x_n \right)^2.$$

**Total objective.** The training objective is a weighted sum of the above components:

$$\mathcal{L}_{\mathrm{total}} = \mathcal{L}_{\mathrm{fape}} + \mathcal{L}_{\mathrm{lddt}} + 0.02\mathcal{L}_{\mathrm{viol}} + \mathcal{L}_{\mathrm{srcv}} + 4\mathcal{L}_{\mathrm{mse}} - \mathcal{L}_{\mathrm{affinity}}$$

## A.6 BIND AFFINITY PREDICTION MODEL

**Squeeze-and-Excitation (SE) for Topological–Spectral Features.** We use squeeze-and-excitation (SE) channel attention to adaptively reweight multi-channel topological–spectral features before prediction or reward computation. Given a feature tensor $X \in \mathbb{R}^{B \times C \times D_1 \times \cdots \times D_m}$, SE first performs a permutation-invariant squeeze by global averaging over non-channel axes to obtain $z \in \mathbb{R}^{B \times C}$. An excitation MLP with bottleneck ratio $r$ then produces per-channel gates $s = \sigma(W_2 \phi(W_1 z)) \in (0,1)^{B \times C}$, which rescale the original channels via $\widetilde{X}_{:,c,\cdot} = s_{:,c} X_{:,c,\cdot}$ (typically within a residual path for stability). This content-dependent modulation introduces negligible overhead ($\approx 2C^2/r$ parameters) yet provides global, sample-specific channel importances. For topological–spectral inputs, SE is advantageous because it mitigates signal cancellation across signed statistics and filtration scales, adapts to variability in interface size and composition, and preserves the symmetry properties of spectral summaries through invariant pooling. Practically, we apply SE before collapsing scales so the gate sees full multi-scale context, and for any signed statistic $x$ we use a sign-split representation $(x^+, x^-)$ to allow independent modulation of positive and negative evidence. When the downstream scorer is kept frozen during fine-tuning, this reweighting helps align intermediate representations with the scorer's preferred basis, improving robustness and affinity correlation with minimal architectural complexity.

**Hybrid Projections** We adopt a hybrid representation that combines a low-rank bilinear projection with a residual full projection to capture cross-channel interactions without incurring quadratic cost while preserving full expressivity. Given two feature vectors $a \in \mathbb{R}^p$ and $b \in \mathbb{R}^q$ (e.g., antigen/antibody, scale/statistic), a full bilinear map uses $a^\top W b$ with $W \in \mathbb{R}^{p \times q}$. We approximate $W$ by rank-$r$ factors $U \in \mathbb{R}^{p \times r}$, $V \in \mathbb{R}^{q \times r}$ and define

$$h_{\mathrm{bil}} = \Phi\big((U^\top a) \odot (V^\top b)\big) \in \mathbb{R}^d,$$

where $\odot$ is elementwise product and $\Phi : \mathbb{R}^r \to \mathbb{R}^d$ is a small MLP or linear head. In parallel, a residual full projection aggregates first-order information,

$$h_{\mathrm{res}} = W_a a + W_b b + b_0, \qquad W_a \in \mathbb{R}^{d \times p}, \, W_b \in \mathbb{R}^{d \times q},$$

and the hybrid feature is $h = h_{\mathrm{res}} + h_{\mathrm{bil}}$. This design captures second-order interactions through the low-rank bilinear branch with $O(r(p + q))$ parameters while the residual branch ensures gradient flow, stabilizes training, and recovers full linear expressivity when interactions are weak. In practice we use small $r$ (e.g., 8–64), apply normalization before the branches, and optionally gate the bilinear term with a sigmoid or softplus scalar to prevent dominance early in training.

**Transformers** We employ a Transformer(Vaswani et al., 2017) with multi-head self-attention to aggregate and mix information across channels, scales, and interface regions. Self-attention provides content-adaptive weighting among tokens, enabling the model to capture long-range dependencies and nonlocal couplings that are difficult for fixed receptive-field operators. Multi-head attention decomposes this process into parallel subspaces, so distinct heads can specialize in complementary interaction patterns (e.g., hydrophobic vs. polar cues, short- vs. long-range scales, or antigen vs. antibody roles), improving expressivity without incurring a prohibitive parameter cost. In our setting, representing topological–spectral descriptors as a set of tokens (across channel pairs and filtration radii) allows the Transformer to (i) perform permutation-invariant set aggregation with learned, data-dependent weights; (ii) selectively emphasize salient channels and scales while suppressing distracting ones, mitigating signal cancellation; and (iii) fuse heterogeneous cues through cross-token mixing that is more flexible than hard-coded pooling. Relative positional or geometric

encodings (e.g., functions of inter-token scale gaps or interface geometry) can be injected to guide attention with physically meaningful priors. Combined with SE reweighting and the hybrid bilinear–residual projection, the Transformer serves as a versatile, interpretable aggregator that boosts downstream affinity correlation and stability with modest computational overhead.

**Training Details**   Binding affinity prediction model of TopoScorer is trained on PPB-Affinity(Liu et al., 2024) Dataset—the largest publicly available dataset of protein–protein binding affinities. From its PDBbind v2020(Liu et al., 2015), SAbDab(Dunbar et al., 2014), and Affinity Benchmark v5.5(Vreven et al., 2015) components, we preprocess entries by splitting each complex into individual protein-protein interfaces according to the participating chain identifiers; interfaces derived from the same PDB entry share the same affinity label. After duplicate removal, this yields 4,818 labeled interfaces. For SKEMPI v2.0, we use FoldX to construct mutant complex structures from the corresponding wild-type templates, guided by the annotated mutation sites. We adopt the reported $K_D$ as the affinity measurement and convert it to a regression target via the standard transformation $-\log K_D$. For each interface, we extract multi-scale topological spectral features from the atomic coordinates and train the model to predict affinity from these features using mean-squared error (MSE) as training loss. We reserve 474 interfaces released after June 30, 2018 as a held-out test set, split the remainder into training and validation sets at a 7:3 ratio. To prevent potential data leak from similar interfaces, we adopted the interface-similarity protocol recommended in (Bushuiev et al., 2024b). Concretely, following Bushuiev et al. (Bushuiev et al., 2024a), we extracted PPI interfaces for all complexes using 6 Å heavy-atom contacts between the two partners (as in PPIRef) and embedded all interfaces with the iDist algorithm and, for each test interface, computed its iDist distance to all training interfaces. We identified near-duplicates as test interfaces having at least one training interface with iDist distance less than 0.04, which is reported to correspond to near-duplicate 6 Å interfaces. Finally, we removed these near-duplicate test entries (i.e., potential leaks) and re-evaluated all baselines and TopoScorer on the resulting leakage-controlled benchmark. After data leak filter, there are 351 remaining complex. The model was trained on 4 A800 GPUs for roughly two days until either reaching the maximum number of iterations or the validation loss stops decreasing; the final model is selected by the best validation performance.

For mutation task, we trained TopoScorer with data from SKEMPI v2.0. We generate structures of mutations by FoldX(BuildModel) and obtained 5550 mutation complexes with affinity labels. Our single and multiple mutations test sets are curated from commonly used benchmark S1131(Xiong et al., 2017) and M1707(Zhang et al., 2020) using the same method as above to prevent data leak, containing 1067 and 782 complexes, respectively. We split the training set into training and validation sets at a 7:3 ratio, and train on 4 A800 GPUs for 6 hours on 4 A800 GPUs. Parameter settings for affinity prediction model and for PTHL feature extraction are in Table. 3 and Table.4

Table 3: Hyperparameters and defaults for `AffinityScaleTransformer`.

| Hyperparameter | Meaning | Default |
|---|---|---|
| m | number of statistics channels across scales | 6 |
| c | number of element channels | 143 |
| d_model | token embedding width | 384 |
| d_mid | intermediate width in factorized projection | 192 |
| depth | number of Transformer blocks | 6 |
| nhead | attention heads per block | 8 |
| mlp_ratio | MLP expansion ratio in blocks | 4.0 |
| dropout | global dropout rate | 0.10 |
| max_len | maximum sequence length for positional encoding | 256 |
| use_cls_token | prepend a [CLS] token? | True |
| learning rate | base learning rate | $8e-5$ |
| seed | random seed | 12345 |
| warmup_steps | warmup steps | 0.1 |
| max_steps | max training step | 10000 |

Table 4: Hyperparameters for PTHLs feature extraction.

| Name | Type | Default | Meaning / Notes |
|------|------|---------|-----------------|
| device | torch.device | *None* | Computation device (e.g., cuda:0 or cpu). |
| dtype | torch.dtype | torch.float32 | Floating precision for kernels and spectra. |
| eps | float | $1\times10^{-7}$ | Numerical jitter for stability (e.g., inverses, norms). |
| sigma_zero | float | $1\times10^{-5}$ | Width of soft zero-indicator; smaller $\Rightarrow$ sharper near zero. |
| tau | float | 0.05 | Temperature for soft-min/max (and soft gates); smaller $\Rightarrow$ closer to hard extremum. |
| alpha | float | 0.01 Å | Threshold softness (in Å); smaller $\Rightarrow$ closer to a hard distance threshold. |
| consider_field | float | 10 Å | Neighborhood selection radius: include atoms whose distance to *any* atom in the opposite protein is < this value. |
| dis_cut_off | float | 10 Å | Maximum filtration radius (upper bound of the distance threshold sweep). |
| interval | float | 0.1 Å | Filtration step size (increment of the distance threshold). |

## A.7   BASELINE MODELS

**Rosetta**   We used the Rosetta molecular modeling suite via its Python interface(Chaudhury et al., 2010). All SKEMPI2 complex structures were first minimized with the relax protocol. Mutants were then generated using the Cartesian-space mutation workflow cartesian_ddg under the ref2015_cart energy function (Rosetta v2023.49). For each protein coomplex, interfacial binding energies were estimated with InterfaceAnalyzer metrics (dG_separated and dSASAx100). For mutation effect evaluation, the mutation-induced change in binding free energy was computed as $\Delta\Delta G_{\text{bind}} = \Delta G_{\text{bind}}(\text{mutant}) - \Delta G_{\text{bind}}(\text{wild type})$. This pipeline provides a standard, reproducible Rosetta estimate of mutation effects on protein–protein affinity.

**FoldX**   FoldX(Delgado et al., 2019; Schymkowitz et al., 2005) is a fast, empirical energy function for proteins that explicitly models van der Waals, hydrogen bonding, electrostatics, solvation/hydrophobic effects, and entropic terms (e.g., side–chain and backbone contributions). In our setup, each SKEMPI complex was first standardized with RepairPDB, after which mutant structures were generated using BuildModel. Binding energies for wild type and mutants were then evaluated with AnalyseComplex, and the mutation-induced change in binding free energy was reported as $\Delta\Delta G_{\text{bind}} = \Delta G_{\text{bind}}(\text{mutant}) - \Delta G_{\text{bind}}(\text{wild type})$. This pipeline provides a rapid and robust baseline for high-throughput mutation scoring and interface optimization with FoldX.

**ESM-1v**   ESM-1v(Meier et al., 2021a) is a sequence-only protein language model trained at scale and used for zero-shot variant effect prediction: given a wild-type sequence, it assigns likelihoods to single or multiple substitutions and scores functional impact via log-likelihood (or log-odds) differences without task-specific supervision (Meier et al., 2021a). As a general-purpose, structure-agnostic baseline, ESM-1v has proved competitive across diverse mutational assays and complements structure-conditioned design models by providing fast, alignment-free estimates of mutational tolerance.

**ESM-IF**   ESM-IF(Hsu et al., 2022b) is a structure-conditioned protein language model that, given a protein backbone, predicts sequences compatible with that structure and assigns conditional log-likelihoods to any provided sequence. In design and evaluation settings, it can thus generate or rescore candidates (including interface binders). In our use, we compute per-residue conditional log-likelihoods on the wild type and mutant backbones and aggregate their differences over mutated (or interfacial) sites as a proxy signal for affinity or $\Delta\Delta G_{\text{bind}}$.

**PRODIGY**    PRODIGY(Xue et al., 2016) is a structure-based baseline for predicting protein–protein binding affinity from a given complex structure . It uses simple, interpretable interfacial descriptors—principally the number and types of interfacial contacts plus noninteracting surface (NIS) properties—within a linear model to estimate affinity (typically reported as $\Delta G$ in kcal/mol or converted to $K_D$ at standard temperature). Inputs are the 3D coordinates of a docked or experimentally determined complex; no training or fine-tuning is required at inference. In our experiments, we use PRODIGY "as is" to score predicted complexes as a classical baseline. Typical limitations include sensitivity to interface delineation and pose quality (e.g., suboptimal docking poses or incomplete interfaces), and the method is not differentiable, so it cannot provide gradients to upstream generative models.

**DSMBind**    DSMBind(Jin et al., 2023) adopts an energy-based, SE(3)-equivariant denoising score-matching framework to learn a continuous "energy landscape" (score field) over protein–protein interactions without explicit supervision on binding energies. The learned score provides a versatile signal for ranking affinity, assessing docked poses, and guiding binder (e.g., nanobody) design; we use its energy/score outputs to compare mutants and complexes.

**DDGPred**    DDGPred(Shan et al., 2022) denotes supervised deep regressors for mutation-induced changes in protein–protein binding free energy, typically trained on curated $\Delta\Delta G$ datasets. Inputs combine complex structures with localized geometric/energetic descriptors around the mutation site, and the model outputs $\Delta\Delta G_{\text{bind}}$ for single or multiple point mutations. We include a representative DDGPred implementation as a learning-based baseline alongside physics-based methods (Rosetta/FoldX).

**RDE-Network**    RDE-Network(Luo et al., 2023) is built upon a Rotamer Density Estimator that learns side-chain rotamer distributions in an unsupervised manner to capture conformational flexibility and entropic effects. Downstream networks map these RDE-derived features to $\Delta\Delta G_{\text{bind}}$, leveraging changes in conformational freedom to explain mutation impacts while reducing reliance on labeled free-energy data.

**TopoNetTree**    TopoNetTree(Cang & Wei, 2017) is a classic persistent-homology model that combines ESPH with CNNs. It starts from topological representations, TopologyNet relies on Betti-number–based persistent homology barcodes as features. It constructs multi-scale topological descriptors around the mutation site using persistent homology and feeds these handcrafted features into a tree-based regression model to predict $\Delta\Delta G$.

**GearBind**    GearBind(Cai et al., 2024) is a pretrainable geometric graph neural network for protein–protein binding affinity change ($\Delta\Delta G$) prediction. It is pretrained on CATH using contrastive learning and fine-tuned on SKEMPI with a regression loss. Here we provide the inference code of GearBind.

## A.8    Additional Experiments

### A.8.1    Stree Test of TopoScorer's Sensitivity

In addition to using FoldX for mutant structures, we have explicitly stress-tested TopoScorer's sensitivity to coordinate perturbations. Concretely, for all complexes in our test set of binding affinity prediction, we added isotropic coordinate noise of different magnitudes (0.1–1.0 Å) to the atomic coordinates and recomputed the predicted binding affinity. We then measured the mean relative change in the predictions across all mutants; the results are summarized in Table 5.

These results show that TopoScorer is highly stable under realistic levels of structural noise: small perturbations ($\leq$ 0.5 Å) induce less than 5% change on average, and even sizeable perturbations on the order of 1.0 Å (comparable to typical AF-style backbone deviations) only lead to $\approx$ 10% variations, indicating that TopoScorer's conclusions are robust to moderate coordinate jitter and side-chain positioning noise.

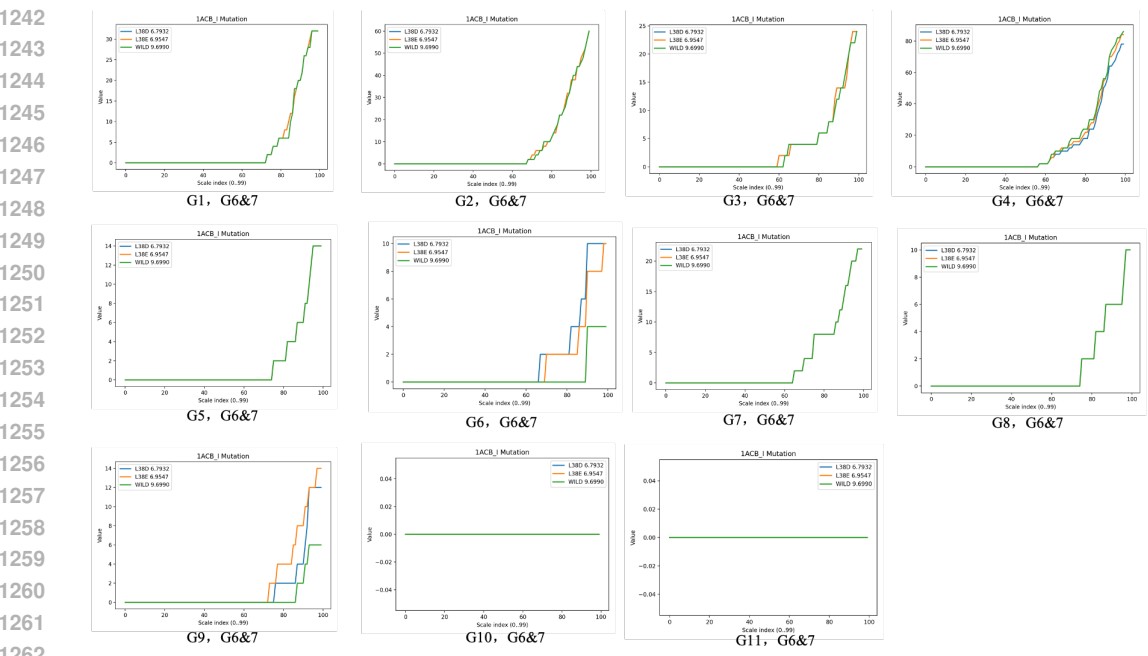

Figure 3: Sum of eigenvalues of PTHLs between ligand glutamate/aspartate carboxylate oxygens and each receptor channel.

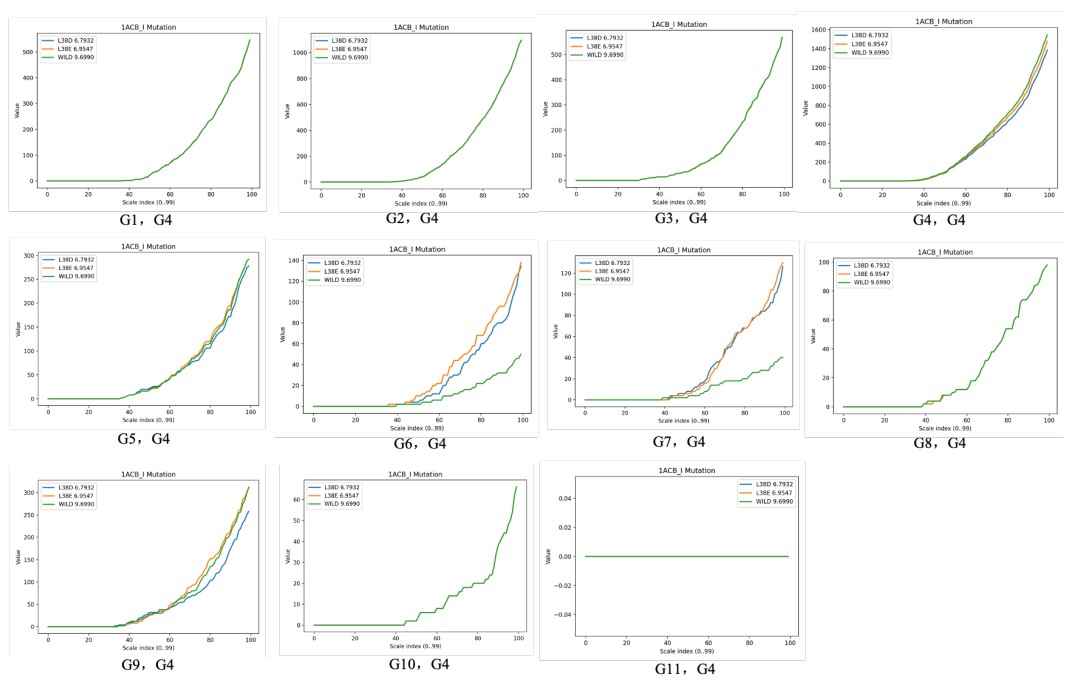

Figure 4: Sum of eigenvalues of PTHLs between ligand aliphatic side-chain carbon and each receptor channel.

### A.8.2 BOND COUNT FEATURE ANALYSIS

Fig 5 visualizes the bond count curves for 1ACB and the two LI38 variants (LI38D and LI38E) across several interaction channels. For each system, the bond count increases with the distance

Table 5: Sensitivity of TopoScorer to isotropic coordinate noise on the binding-affinity test set. Values report the mean relative change in predicted affinity across all mutants.

| Noise level (Å) | Mean change (%) |
| --- | --- |
| 0.1 | 1.13 |
| 0.2 | 3.95 |
| 0.5 | 4.56 |
| 1.0 | 10.76 |

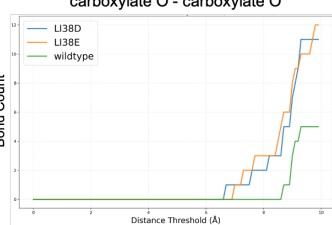 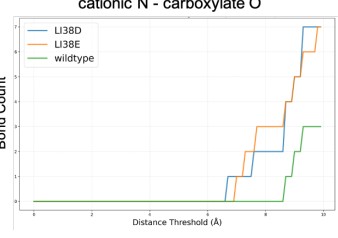 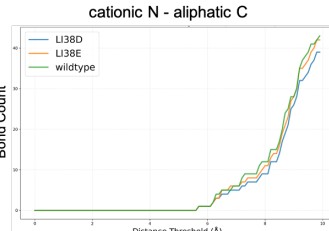

Figure 5: Bond count diagram between carboxylate O - carboxylate O , cationic N - carboxylate O, and cationic N - aliphatic C of 1ACB and its mutants.

threshold, qualitatively mirroring the monotonic trend we observe for our spectral topological feature $\lambda_{\text{sum}}$. However, the bond count curves are clearly much noisier: the trajectories for wild type and the two mutants frequently cross, and small local fluctuations in contact number obscure a consistent ordering between variants. In contrast, $\lambda_{\text{sum}}$ aggregates information from the full Laplacian spectrum of each interaction graph, capturing the overall strength and organization of contacts rather than just their raw counts. As a result, the $\lambda_{\text{sum}}$ curves are smoother, show fewer crossings, and separate the three complexes more robustly across scales. This reduced noise and improved discriminability help explain why models based on simple bond count features underperform those built on our spectral topological descriptors.

### A.8.3    MORE INTERPRETABILITY CASE STUDIES

As shown in Fig 6, we analysed the zero–eigenvalue count of the element-specific graph Laplacian, which reports the number of connected components in each interaction subgraph as the distance threshold increases. In Fig 6(b), which corresponds to the anion–anion subgraph between receptor carboxylate oxygens and ligand carboxylate oxygens, the wild type exhibits the largest zero counts across all radii, while the E79A and E79A_K80A mutants show progressively lower curves. This indicates that the wild-type interface contains multiple disjoint anionic clusters around E79, reflecting a highly fragmented and electrostatically frustrated acidic patch; removal of E79 collapses parts of this network and topologically simplifies the anionic environment, in line with the observed increase in affinity. Fig 6(c), which tracks the cation–anion subgraph between receptor nitrogens and ligand carboxylate oxygens, shows a similar pattern: the wild type has the highest number of connected components, suggesting an over-structured but fragmented Lys–Asp/Glu network ("electrostatic cage"), whereas the mutants display fewer components, consistent with pruning of suboptimal or partially desolvated salt-bridge configurations while retaining the most productive ones. Finally, in Fig 6(d) , corresponding to interactions between receptor aliphatic carbons and ligand cationic nitrogens, shows only subtle shifts but again follows the affinity trend: the double mutant displays a modest reduction in zero counts relative to the wild type, consistent with replacing K80 by alanine and thereby converting a heterogeneous hydrophobic–cationic environment into a more homogeneous hydrophobic patch. Overall, the downward shifts in the Laplacian zero counts indicate a topological simplification of problematic anionic and cationic subnetworks, which correlates with relief of electrostatic frustration and the stepwise gain in binding affinity from wild type to E79A and to the E79A_K80A double mutant.

As shown in Fig 7, for the three spectral channels that show the clearest correlation with the experimental affinities curves consistently follow the order wild type > R167K > R167N across distance

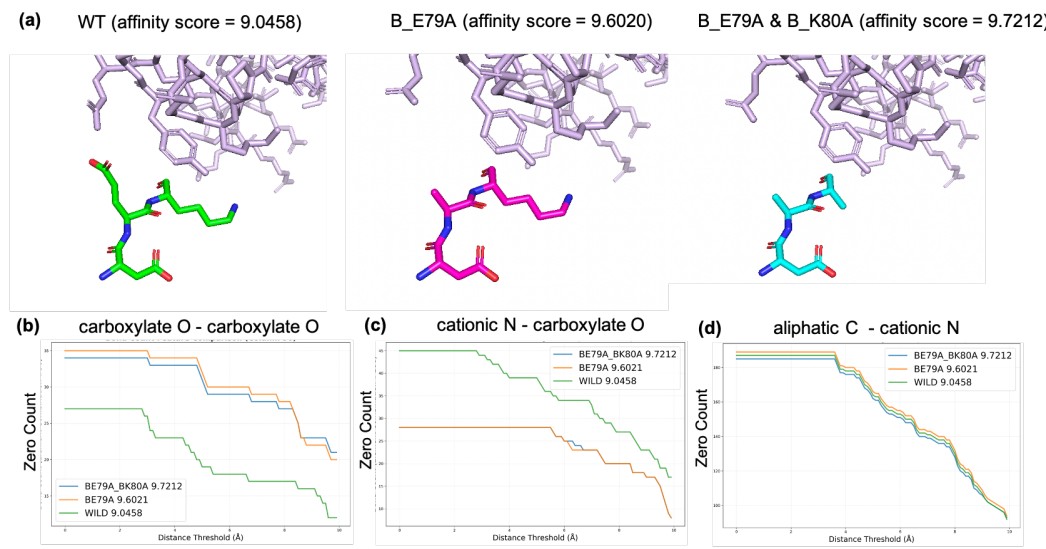

Figure 6: (a) Structures of 1A22 and mutants residues 78–80 (green: wild type ligand chain B; pink: mutant E79A ligand chain B; blue: mutant E79A_K80A ligand chain B; light purple: receptor chain E). (b) Zero count between receptor carboxylate oxygens and ligand carboxylate oxygens. (c) Zero count between receptor cationic nitrogens and ligand carboxylate oxygens. (d) Zero count between receptor aliphatic carbons and ligand cationic nitrogens.

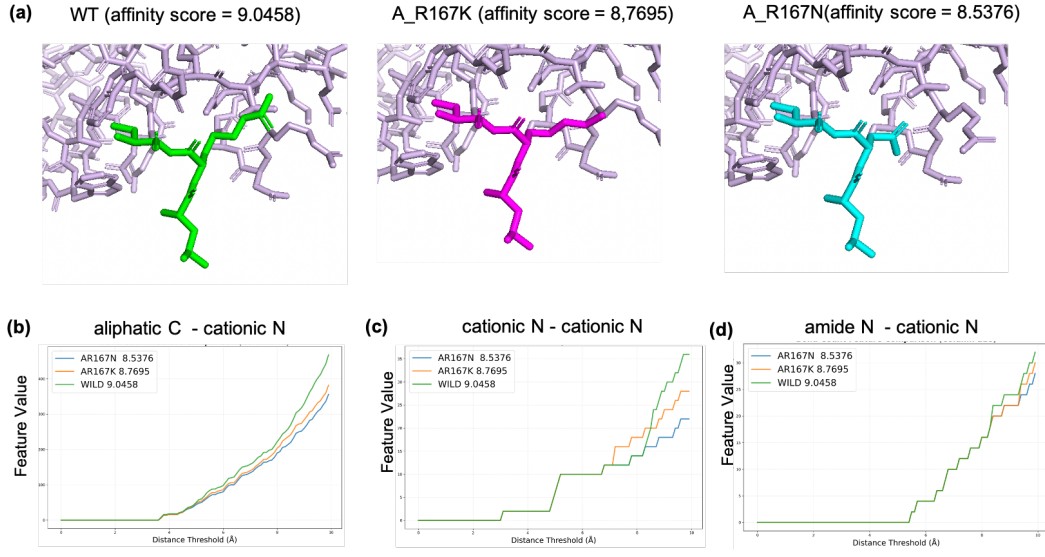

Figure 7: (a) Structures of 1A22 and mutants residues 166–168 (green: wild type ligand chain B; pink: mutant R167K ligand chain A; blue: mutant R167N ligand chain A; light purple: receptor chain E). (b) $\lambda_{sum}$ between receptor aliphatic carbons and ligand cationic nitrogens. (c) $\lambda_{sum}$ between receptor cationic nitrogens and ligand cationic nitrogens. (d) $\lambda_{sum}$ between receptor amide nitrogens and ligand cationic nitrogens.

thresholds. Fig. 7(b) corresponds to the protein aliphatic carbon–ligand cationic nitrogen channel, reporting how well the positively charged group on the ligand is embedded in a hydrophobic shell; the larger $\lambda_{sum}$ values for the wild type indicate a more extensive and coherent hydrophobic–cationic packing environment than in either mutant. Fig. 7(c) captures the protein cationic nitrogen–ligand cationic nitrogen channel and reflects the organization of the interfacial cationic network. Here again

| Class | Atoms (Atom37 names) | Physicochemical meaning |
|---|---|---|
| G1 | N | Backbone amide N; H-bond donor; defines peptide directionality. |
| G2 | CA, C | Backbone $\alpha$-carbon and carbonyl C; main-chain scaffold and geometry. |
| G3 | O | Backbone carbonyl O; H-bond acceptor; drives secondary-structure H-bonds. |
| G4 | CB, CG1, CG2, CD, CE, CG, CD1, CD2 | Aliphatic side-chain C; hydrophobic packing, shape complementarity, van der Waals contacts. |
| G5 | CE1, CE2, CE3, CZ, CZ2, CZ3, CH2 | Aromatic/conjugated ring C (Phe/Tyr/Trp/His); $\pi$-stacking, cation–$\pi$, polarizability. |
| G6 | OD1, OD2 | Aspartate carboxylate O; negatively charged; salt bridges to lysine/arginine; H-bond acceptor. |
| G7 | OE1, OE2 | Glutamate carboxylate O (longer reach than Asp); negatively charged; salt bridges; H-bond acceptor. |
| G8 | OG, OH, OG1 | Serine/threonine/tyrosine hydroxyl O; donor/acceptor; interfacial polarity. |
| G9 | NE, NH1, NH2, NZ | Cationic N (arginine guanidinium NE/NH1/NH2; lysine NZ); salt bridges; H-bond donor. |
| G10 | ND2, NE2, ND1, NE1 | Amide/imidazole N (asparagine ND2, glutamine NE2, histidine ND1/NE2, tryptophan NE1); pH-dependent donor/acceptor (His), polar contacts. |
| G11 | SE, SD | Chalcogen in selenomethionine (SE) and sulfur in methionine (SD); soft, polarizable; thioether/selenoether contacts. |

Table 6: Atom37 groups (G1–G11) by physicochemical role.

the wild type shows the strongest spectral signature, consistent with a more structured positive-charge network that helps position and stabilize the ligand, while R167K and especially R167N progressively disrupt this network. Finally, Fig. 7(d) corresponds to the protein amide nitrogen–ligand cationic nitrogen channel, highlighting a polar scaffold that supports the ligand's charged group; the monotonic decrease of $\lambda_{\text{sum}}$ from wild type to R167K to R167N suggests a stepwise loss of this polar support. Together, these three indices indicate that mutations at R167 primarily weaken a cooperative hydrophobic–cationic–polar interaction network at the interface, in line with the observed reduction in binding affinity.

A.9    PHYSICOCHEMICAL-ROLE–AWARE CLASSES

Physicochemical-role–aware classes partitions and explanations are presented in Table 6.

