# OpenReview forum: "TopoScorer: a light, interpretable predictor for protein-protein binding affinity"
_ICLR.cc/2026/Conference — Submitted to ICLR 2026_

### Official Review · Reviewer_sgQZ · 2025-10-16

**Soundness:** 2
**Presentation:** 2
**Contribution:** 1
**Rating:** 2
**Confidence:** 4

**Summary:**

This paper introduces TopoScorer, a model designed to predict the binding affinity of protein-protein interactions. The method is evaluated using standard data, and it also includes a case study demonstrating how an antibody generative model can be guided to produce designs with higher affinity. While the method is presented as being lightweight, interpretable, and end-to-end trainable, each of these aspects currently exhibits significant weaknesses.

**Strengths:**

- The featurization and model architecture are described with great detail.
- The concept of fine-tuning a generative antibody model to enhance affinity is both practically interesting and innovative.

**Weaknesses:**

**Major**

- Although the featurization and model architecture are well-explained, there is a lack of clarity regarding the training details, loss function, and data. Section 3.3, "BINDING AFFINITY PREDICTION MODEL," only covers the "Model Architecture" paragraph, leaving the training objective and data unclear.
- The proposed method appears to underperform existing approaches. For scoring mutations, Table 1 shows only a slight improvement in Spearman correlation over RDE-Net, while its Pearson correlation is substantially lower. A similar trend is observed for affinity prediction in Table 1 when compared against DSMBind, a fully unsupervised method.
- The evaluation seems to be affected by data leakage, and further analysis is needed. The affinity prediction section (Section 4.1) does not specify how the complexes were split (e.g., sequence or structure similarity). The data for antibody design is split by release time (Section 4.2), which is known to cause data leakage when used as the sole criterion [1].
- The method's key novelties, being "lightweight, interpretable, and end-to-end trainable", have considerable issues:
  - The claim that the model is "lightweight" is not supported. Running time is not analyzed. Section 4.1 only states that TopoScore has  approximately four times fewer parameters than RDE-Network, but it remains unconfirmed whether this translates into substantially lower memory or time consumption.
  - The "interpretability" of the method is only demonstrated through a single example (Figure 2b). It is unclear whether similar interpretability could be achieved by simply calculating the number of bonds rather than relying on topological features (please refer to Question 1).
  - While the Abstract states that "existing deep learning based approaches lack interpretability and a differentiable path from affinity back to the interface," multiple prior methods are indeed end-to-end differentiable, including for example [2], and DDGPred, which is mentioned in Related Work as end-to-end differentiable (line 94).

**Minor**
- The paper does not cite or discuss TopNetTree, a previous method that also uses topological features of protein-protein interfaces to predict affinity [3].
- Some sentences are missing references:
  - [Line 39]. The sentence "Deep learning has become the dominant paradigm for protein–protein binding affinity prediction, delivering state-of-the-art accuracy and throughput" lacks a reference.
  - [Line 45]. The "High training cost" of prior methods is not supported by any reference.
  - [Line 106]. "Reviews" are mentioned, but only one is cited.

References

- [1] Bushuiev et al, 2024, “Revealing data leakage in protein interaction benchmarks”, https://arxiv.org/abs/2404.10457
- [2] Shuai et al, 2025, “Sidechain conditioning and modeling for full-atom protein sequence design with FAMPNN”, https://www.biorxiv.org/content/10.1101/2025.02.13.637498v1.full.pdf
- [3] Want et al, 2020, “A topology-based network tree for the prediction of protein–protein binding affinity changes following mutation”, https://www.nature.com/articles/s42256-020-0149-6

**Questions:**

1. What would be the outcome if Figures 2e, f, and g were replaced with simple bar plots showing the counts of respective bonds? For instance, instead of Figure 2e, if there were simply three numbers representing carboxylate O - carboxylate O bonds for each of the three structures, would the same trend still be observable?

---

> ### Author Response · Authors · 2025-11-22
>
> We sincerely thank the reviewer for your detailed comments about performance, data leakage, lightweight claims, model details, and interpretability. We would like to address them to our knowledge.
>
> **W1: Although the featurization and model architecture are well-explained, there is a lack of clarity regarding the training details, loss function, and data. Section 3.3, "BINDING AFFINITY PREDICTION MODEL," only covers the "Model Architecture" paragraph, leaving the training objective and data unclear.**
>
> Sorry for the inconvenience but we would like to clarify that these details were **previously included in the initial submission:**
>
> - **Appendix A.5** provides the **full training objectives** for the fine-tuning procedure, with explicit equations, the role of each loss term, training time and devices.
> - **Appendix A.6** provides the binding affinity prediction model’s full training specification, including the datasets/splits, hyperparameters, model settings and the computational setup.
>
> There are links at the last lines in Sec 3.3&3.4 directing to A.5–A.6  for your kind reference.
>
> **W2: The proposed method appears to underperform existing approaches. For scoring mutations, Table 1 shows only a slight improvement in Spearman correlation over RDE-Net, while its Pearson correlation is substantially lower. A similar trend is observed for affinity prediction in Table 1 when compared against DSMBind, a fully unsupervised method.**
>
> Model performance is a central aspect that we carefully monitor and continuously seek to improve. Since the initial submission, we have substantially strengthened the training and evaluation pipeline, including (i) scaling up and re-curating the training data, (ii) a more thorough hyperparameter search (learning rate, optimizer, weight decay, batch size, PTHL temperature parameters, radius grid, etc.), (iii) clustering highly similar interactions in the training set to reduce redundancy and stabilize optimization.
>
> We initially constructed all splits purely based on release time, which, as pointed out by Bushuiev et al. [1], can still allow substantial leakage when used as the only criterion. To address this, we revisited all our splits and adopted the interface-similarity protocol recommended in Bushuiev et al. [1,2]. We removed these near-duplicate test entries (i.e., potential leaks) and re-evaluated all baselines and TopoScorer on the resulting leakage-controlled benchmark(**See our response to W3**). Importantly, after removing all potential near-duplicate interfaces, our model still **achieves** **state-of-the-art performance in** **both** **Spearman and Pearson** on  **binding affinity prediction**, as well as **SOTA Spearman correlations and second best Pearson on single-mutation and multi-mutation subsets.** The updated results are reported in the revised Table 1.
>
> | Method       | Affinity Spearman | Affinity Pearson | Single-mutation Spearman | Single-mutation Pearson | Multi-mutations Spearman | Multi-mutations Pearson |
> |-------------|-------------------|------------------|--------------------------|-------------------------|--------------------------|-------------------------|
> | PyRosetta   | 0.1856            | 0.1954           | 0.3422                   | 0.3285                  | 0.2927                   | 0.2258                  |
> | FoldX       | *0.3295*     | 0.3008    | 0.4355                   | 0.4586                  | 0.3734                   | 0.3241                  |
> | ESM-1v      | 0.1034            | 0.0876           | 0.1524                   | 0.1921                  | 0.1512                   | 0.1736                  |
> | ESM-IF      | 0.0530            | 0.0244           | 0.1116                   | 0.1047                  | 0.1697                   | 0.0700                  |
> | PRODIGY     | 0.1549            | 0.1277           | 0.3233                   | 0.2902                  | 0.3421                   | 0.3236                  |
> | DSMBind     | 0.3072            | *0.3269*           | 0.3530                   | 0.3261                  | 0.3673                   | 0.2954                  |
> | DDGPred     | —                 | —                | *0.5522*          | 0.5303                  | 0.4585                   | 0.5638                  |
> | RDE-Network | —                 | —                | 0.5127                   | **0.6067**              | 0.5397                   | **0.6108**              |
> | GearBind    | —                 | —                | 0.5014                   | 0.5496                  | *0.5470*          | 0.5616                  |
> | TopoNetTree | —                 | —                | 0.5185                   | 0.5508                  | —                        | —                       |
> | TopoScorer  | **0.3848**        | **0.3804**       | **0.5876**               | *0.5615*          | **0.5704**               | *0.5652*           |

---

> ### Author Response · Authors · 2025-11-22
>
> We would like to clarify that Spearman and Pearson reflects the fact that they quantify different aspects of model behavior: Spearman measures ranking quality, focusing on whether higher-affinity complexes are consistently assigned higher scores, while Pearson measures linear calibration of the predicted values against ground truth: it is affected by global scaling, offset, and a small number of outliers.
>
> From a practical application perspective, this distinction is particularly important: current affinity prediction models (including ours and prior work) are still far from perfectly accurate in terms of absolute ΔΔG values. As a result, most downstream tasks in protein and PPI design (e.g., virtual screening, prioritizing mutants, guiding generative models) place much more emphasis on **reliable ranking** of candidates rather than on the exact numerical value of the predicted affinity. In these scenarios, a model that provides a stable and accurate ordering of candidates (high Spearman) is typically more useful in practice than one that achieves slightly better calibration of absolute values but weaker ranking performance. Additionally, prior evaluations of Boltz-2[3] show that affinity prediction performance varies markedly across assays and can be confounded by errors in predicted structures, limited generalization to unseen protein families, and sensitivity to out-of-distribution small molecules; under such variability, **ranking accuracy (Spearman) is typically the more robust indicator of practical utility**. Pearson remains a useful secondary measure of how well the score can approximate absolute affinities on a given dataset, and we report it for completeness. In the revised version, after improving the training pipeline and removing potential data leakage, TopoScorer now achieves stronger performance on both Spearman and Pearson (see updated Table 1). Nevertheless, even in settings where another method has a marginally higher Pearson while TopoScorer has substantially higher Spearman, it’s rather a **trade-off** between ranking accuracy and calibration, with ranking being the primary objective in most affinity-based design workflows.
>
> Crucially, we would like to emphasize that **Spearman and Pearson are not the only criteria** we optimize for. A main motivation of TopoScorer is to provide a practical scoring module that is not only accurate but also **lightweight, fast, differentiable, and interpretable**. As detailed in our revision, TopoScorer has substantially fewer parameters and achieves **fast and stable inference time**; its PTHL-based representation is **fully differentiable** with respect to atomic coordinates, enabling direct gradient-based integration into generative models; and its **spectral features** admit meaningful attributions that go beyond black-box predictions. We believe these additional properties are highly relevant for real-world PPI design workflows, and they complement the strong Spearman/Pearson performance.
>
> Compared with RDE-Network, which obtains slightly better Pearson scores on mutation tasks, our method offers a favorable  strength in efficiency, parameter budget and interpretability: (i)TopoScorer achieves substantially fast and stable inference time($5.01\pm0.1$ ms per sample) than RDE-Network($1383.41\pm 111.18$ ms per sample) (**See our respond to W3**), (ii) TopoScorer uses substantially fewer parameters, as reported in the manuscript, the competing RDE-Network relies on large pretrained components (133M+63M parameters), whereas our model has only ∼43M parameters, (iii) TopoScorer provides intrinsically interpretable, differentiable PTHL-based spectral features that can be directly integrated as a training signal in generative frameworks (e.g., IgGM finetuning), (iv) RDE-Network are specifically designed for ΔΔG prediction, whereas TopoScorer can directly predict absolute binding affinity, giving it a **broader application range** in tasks that require scoring or ranking arbitrary PPI interfaces.
>
> To support our claims, we measured the inference time and discussed time and parameters in our response to W4. And we give further explanation about interpretability in our response to Q1.
>
> - [1] Bushuiev et al, 2024, “Revealing data leakage in protein interaction benchmarks”, https://arxiv.org/abs/2404.10457
> - [2] Bushuiev et al., 2024, Learning to design protein–protein interactions with enhanced generalization, ICLR 2024.
> -  [3]Saro Passaro, Gabriele Corso, Jeremy Wohlwend, Mateo Reveiz, Stephan Thaler, Vignesh Ram
>     Somnath, Noah Getz, Tally Portnoi, Julien Roy, Hannes Stark, David Kwabi-Addo, Dominique
>     Beaini, Tommi Jaakkola, and Regina Barzilay. Boltz-2: Towards accurate and efficient binding
>     affinity prediction. bioRxiv, 2025. doi: 10.1101/2025.06.14.659707. URL [https://doi](https://doi/).
>     org/10.1101/2025.06.14.659707.

---

> ### Author Response · Authors · 2025-11-22
>
> **W3: The evaluation seems to be affected by data leakage, and further analysis is needed. The affinity prediction section (Section 4.1) does not specify how the complexes were split (e.g., sequence or structure similarity). The data for antibody design is split by release time (Section 4.2), which is known to cause data leakage when used as the sole criterion**
>
> We appreciate the reviewer’s concern about potential data leakage in PPI benchmarks and antibody design tasks. We initially constructed all splits purely based on release time, which, as pointed out by Bushuiev et al. [1], can still allow substantial leakage when used as the only criterion. To address this, we revisited all our splits and adopted the interface-similarity protocol recommended in Bushuiev et al. [1,2]. Concretely, following Bushuiev et al., we:
>
> 1. Extracted PPI interfaces for all complexes using 6 Å heavy-atom contacts between the two partners (as in PPIRef).
> 2. Embedded all interfaces with the iDist algorithm and, for each test interface, computed its iDist distance to *all* training interfaces.
> 3. Identified near-duplicates as test interfaces having at least one training interface with iDist distance ≤ 0.04, which is reported to correspond to near-duplicate 6 Å interfaces in [1,2].
> 4. Removed these near-duplicate test entries (i.e., potential leaks) and re-evaluated all baselines and TopoScorer on the resulting leakage-controlled benchmark.
>
> We have updated the performance tables accordingly. Importantly, after removing all potential near-duplicate interfaces, our model still achieves *state-of-the-art* performance on binding affinity prediction: we obtain the best Spearman and Pearson correlations on the full PPI affinity benchmark, as well as SOTA Spearman and second best correlations on both single-mutation and multi-mutation subsets. This confirms that the improvements of TopoScorer are not due to data leakage, but persist under the stricter, interface-similarity-based evaluation protocol advocated by Bushuiev et al. For transparency and reproducibility, we provide the detailed train–test interface similarity statistics and the leakage-filtered benchmark files (including updated performance tables for all methods) in our anonymous GitHub repository linked in the manuscript.
>
> We clarify in Appendix A.6 that the affinity prediction benchmark is now split using interface similarity (iDist-based) as the primary criterion.
>
> [1] Bushuiev et al., 2024, *Revealing data leakage in protein interaction benchmarks*, arXiv:2404.10457.
>
> [2] Bushuiev et al., 2024, *Learning to design protein–protein interactions with enhanced generalization*, ICLR 2024.

---

> ### Author Response · Authors · 2025-11-22
>
> **W4: The method's key novelties, being "lightweight, interpretable, and end-to-end trainable", have considerable issues**:
> (1) The claim that the model is "lightweight" is not supported. Running time is not analyzed. Section 4.1 only states that TopoScore has approximately four times fewer parameters than RDE-Network, but it remains unconfirmed whether this translates into substantially lower memory or time consumption.
> (2) The "interpretability" of the method is only demonstrated through a single example (Figure 2b). It is unclear whether similar interpretability could be achieved by simply calculating the number of bonds rather than relying on topological features (please refer to Question 1).
> (3) While the Abstract states that "existing deep learning based approaches lack interpretability and a differentiable path from affinity back to the interface," multiple prior methods are indeed end-to-end differentiable, including for example [2], and DDGPred, which is mentioned in Related Work as end-to-end differentiable (line 94).
>
> First of all, we appreciate the opportunity to clarify the scope of our claim, the three aspects “lightweight, interpretable, and end-to-end trainable” are intended as a **combined** contribution rather than three independent claims. In practice there is a strong **trade-off** between these properties: models that are very deep and expressive tend to be heavy and opaque, while interpretable or topology-based models have historically been non-differentiable and hard to integrate into modern pipelines. Therefore, our contribution is **not** that each of “lightweight”, “interpretable”, and “end-to-end trainable” is entirely novel **in isolation,** but that TopoScorer **combines all three properties** in a single, elegant, PTHL-based PPI affinity model that (i) is lightweight in parameters and runtime, (ii) exposes physically meaningful, PTHL-based structure attributions, and (iii) is fully differentiable from affinity back to atomic coordinates, while maintaining competitive or state-of-the-art performance on the leakage-controlled benchmarks.
>
> We thank the reviewer for the suggestion in (1) and have **added inference-time comparison**. To provide a fair runtime comparison, we additionally measure the inference time per sample on the single-mutation task, using the same cpu. **Results in Fig 2(k).**
>
> TopoScorer achieves remarkably fast and stable inference time($5.01\pm0.1$ ms per sample) among compared models. DSMBind is also computationally efficient, but its predictive accuracy on our benchmarks is substantially lower than TopoScorer, so it does not offer the same balance of speed and reliability. Taken together, TopoScorer strikes a rare and favorable balance between predictive accuracy, parameter efficiency, and inference speed, which is not achieved by the other methods we compare against.
>
> As for the concern (2), we give further explanation about interpretability in our respond to Q1.
>
> (3) We do **not** mean to suggest that no prior affinity model is differentiable—indeed, DDGPred and other deep learning approaches are end-to-end differentiable, and **we have weakened our claim in the paper**. But compared to DDGPred and similar differentiable models, TopoScorer offers a complementary trade-off: it is reliable in performance, fast and robust in runtime, exposes explicit topological–spectral features for interpretation and controlled ablations, and is designed from the outset to serve as a differentiable, interface-level scoring module that can be plugged into downstream design loops. Prior topological approaches (persistent diagrams/barcodes, classical PTHL pipelines) are typically non-differentiable, which prevents their direct use as loss functions or classifier guidance in generative models. On the theoretical side, we derive **new differentiable formulations** for PTHL that enrich the toolbox of topological deep learning, and on the practical side we implement these constructions end-to-end and empirically validate that they are stable and effective in real affinity prediction and design tasks. TopoScorer can be used as a drop-in differentiable reward / loss in IgGM finetuning and other generative frameworks, a use case that is difficult to achieve with non-differentiable topological descriptors or heavier black-box models.

---

> ### Author Response · Authors · 2025-11-22
>
> **Q1: What would be the outcome if Figures 2e, f, and g were replaced with simple bar plots showing the counts of respective bonds? For instance, instead of Figure 2e, if there were simply three numbers representing carboxylate O - carboxylate O bonds for each of the three structures, would the same trend still be observable?**
>
> We thank the reviewer for raising the question of whether our interpretability could be achieved by simply counting bonds instead of using topological features. In our setting, a “carboxylate O–O bond” is not a covalent bond but an interfacial contact between two carboxylate oxygens across the interface. A simple bond-count baseline would thus be implemented by counting the number of such O–O contacts per complex (and analogously for other contact types), without any topological or spectral information. To quantitatively assess this, we added an ablation where we **exactly implement the reviewer’s suggestion**: we discard all PTHL-based spectral features and replace them with **multi-scale, multi-channel bond counts**. Concretely, we keep the original 0–10 Å filter with 0.1 Å spacing and, for each radius and channel, we count the number of interfacial contacts (“bonds”) between the corresponding atom groups, yielding a feature tensor of shape \([1, S, C]\). We then train the same prediction model (only adapting the input shape) and evaluate it on the PPB-Affinity test set.
>
> This “bond-count only” model performs substantially worse than TopoScorer:
>
> - **Bond-count only:** Pearson \(r = 0.2090\), Spearman \($\rho = 0.1252$\).
> - **TopoScorer:** Pearson \(r = 0.3804\), Spearman \($\rho = 0.3848\$).
>
> Thus, simply counting the number of contacts at each scale and channel leads to a **large drop in both Pearson and Spearman.** Intuitively, raw bond counts only encode **how many** contacts of a given type exist, but they do not capture **how these contacts are organized** in the interfacial graph—whether they form connected clusters, bridges, bottlenecks, or are unevenly distributed across the interface. In contrast, our PTHL-based Laplacian spectra (e.g., soft zero-eigenvalue counts, $\lambda_{max}$, $\lambda_{sum}$, mean and variance of the positive spectrum summarize rich geometric and connectivity information across radii and channels, enabling the model to distinguish interfaces with similar counts but very different structural organization. PTHL encodes topological structure (connectivity, cavities) together with physical, chemical, and biological interactions into spectral features across multiple scales. As a result, it is more sensitive to the shape of binding pockets, hydrophobic cavities and polar channels, and stereochemical effects, and provides a more global and richly diverse information than pure bond numbers or local geometric descriptors.
>
> Each of our topological spectral descriptors has a different and complementary topological interpretation, which we have explained in detail in **Appendix A.4**. We have added this ablation experiment to **Sec 4.4** and **Fig 2(i)**.
>
> Moreover, as discussed in our noise-sensitivity analysis in Appendix A.8, these PTHL spectral features are also robust to moderate structural noise: coordinate jitter of 0.1–0.5 Å changes TopoScorer’s predictions by <5% on average. Together with the above ablation, this shows that (i) PTHL features provide information well beyond simple contact counts, and (ii) they do so in a numerically stable way that is suitable for both interpretation and downstream optimization.

---

> ### Author Response · Authors · 2025-11-22
>
> **W5: The paper does not cite or discuss TopNetTree, a previous method that also uses topological features of protein-protein interfaces to predict affinity[1]**
>
> We thank the reviewer for the helpful suggestions on additional baselines. In the revised manuscript, we have added TopoNetTree as another persistent-topology-based affinity predictor to the benchmark.  TopoNetTree is a classic persistent-homology model that combines ESPH with CNNs. Both TopoNetTree and TopoScorer start from topological representations, TopologyNet relies on Betti-number–based persistent homology barcodes as features, whereas TopoScorer uses PTHL and their multi-scale spectral statistics. TopoNetTree was originally designed for single-mutation ΔΔG prediction using classical (non-differentiable) persistent homology. In our updated Table 1, TopoScorer consistently outperforms TopoNetTree in terms of Spearman and Pearson correlation, while also generalizing to broader PPI affinity tasks and offering a fully differentiable PTHL formulation. And we have also added TopoNetTree to out Related Works section.
>
> [1] Want et al, 2020, “A topology-based network tree for the prediction of protein–protein binding affinity changes following mutation”, https://www.nature.com/articles/s42256-020-0149-6
>
> **W6: Some sentences are missing references**
>
> We thank the reviewer for kindly pointing out these missing references. We have revised the manuscript accordingly and added appropriate citations.

---

> > ### Author Response · Authors · 2025-11-27
> >
> > Dear Reviewer,
> >
> > Happy Thanksgiving! We sincerely appreciate your time and efforts in reviewing our manuscript and offering constructive suggestions. As the discussion phase is approaching its end, we would like to kindly confirm whether we have sufficiently addressed your concerns. Should there be any remaining questions requiring further clarification, please do not hesitate to let us know. If you are satisfied with our responses, we would greatly appreciate your consideration in adjusting the evaluation scores accordingly.
> >
> > We look forward to your feedback.
> >
> > Best regards,
> >
> > Anonymous Authors

---

> > > ### Comment · Reviewer_sgQZ · 2025-11-27
> > >
> > > I appreciate the authors’ responses, in particular the updated evaluation setup and improved performance. I have therefore raised my score (2->4).
> > >
> > > Nevertheless, the aspects of the method being lightweight and interpretable still have issues.
> > > - In the case of being lightweight, why is the runtime compared on CPU (“To provide a fair runtime comparison, we additionally measure the inference time per sample on the single-mutation task, using the same cpu.”)?
> > > - In the case of interpretability, it seems that the interpretability is still evaluated on only a single example. Additionally, my Question 1 remains unaddressed. The authors have added new experiments with binding affinity prediction, while the question was addressing interpretability. Furthermore, providing practical guidance for end-users/biologists (not experts in topological methods) on how to use the interpretable features would be beneficial.

---

> ### Author Response · Authors · 2025-11-28
>
> Thank you very much for your detailed comments and for acknowledging our updated evaluation setup, removal of data leakage, and the resulting performance improvements. We also appreciate your decision to raise the score (2→4), and we are happy to further clarify the remaining questions you raised.
>
> **1. In the case of being lightweight, why is the runtime compared on CPU?**
>
> Our choice to report CPU runtimes was mainly driven by the baselines we compare against. In particular, both PyRosetta and FoldX are CPU-based tools without standard GPU implementations in typical use, and in practice they are routinely run on CPU only. To avoid giving an unfair advantage to methods that can leverage GPUs, we therefore measured the per-sample inference time for all methods on the same CPU, which matches the realistic deployment setting for these structure-based baselines.
>
> **2. In the case of interpretability, it seems that the interpretability is still evaluated on only a single example. Additionally, my Question 1 remains unaddressed. The authors have added new experiments with binding affinity prediction, while the question was addressing interpretability. Furthermore, providing practical guidance for end-users/biologists (not experts in topological methods) on how to use the interpretable features would be beneficial.**
>
> We thank the reviewer for emphasizing the importance of interpretability beyond a single example. Following this suggestion, we have **added two additional case studies** in Appendix A.8.3 More Interpretability Case Studies. The **first new case** is 1A22 with single and double mutants B_E79A and B_E79A_K80A. For this system we provide structural illustrations of the mutated residues and plot the zero–count curves as a function of the distance threshold, focusing on three mutation–relevant channels: carboxylate O vs. carboxylate O, cationic N vs. carboxylate O, and aliphatic C vs. cationic N. These channels correspond to acidic clustering, salt-bridge networks and hydrophobic–cationic packing, respectively, and we show how the changes in their topological signatures explain the observed affinity gains. The **second new case** is 1A22 and its mutants A_R167K and A_R167N, where we provide structural views plus λ_sum curves for aliphatic C vs. cationic N, cationic N vs. cationic N, and amide N vs. cationic N. These examples illustrate how our features capture cooperative hydrophobic–cationic–polar networks and how their disruption by mutations leads to reduced affinity. Taken together, the three case studies now **cover a broader range of amino-acid types, physicochemical channels and spectral statistics**, providing a more comprehensive and concrete demonstration of the interpretability of our method. In addition, in the subsequent practical guidance for end-users/biologists we **summarize a more general analysis workflow** that can be applied to **generic samples**.
>
> Regarding the reviewer’s Question 1 (why we use topological features instead of simple bond counts), our original answer relied mainly on ablation results, which may have been insufficiently intuitive. We have now added an explicit comparison in Appendix A.8.2 Bond Count Feature Analysis, using the 1ACB example and its two mutants. For the same chemically meaningful channels (carboxylate O–carboxylate O, cationic N–carboxylate O, and cationic N–aliphatic C), we plot both the bond-count curves and the corresponding λ_sum curves as a function of distance. While bond counts also increase monotonically with the threshold, the curves are much noisier: trajectories for wild type and mutants frequently cross, and small local fluctuations in raw contact numbers obscure a consistent ordering between variants. In contrast, λ_sum aggregates information from the full Laplacian spectrum, capturing the overall strength and organization of the interaction network rather than individual contacts. The resulting curves are smoother, exhibit far fewer crossings, and separate the three complexes more robustly across scales. Combined with the quantitative ablation experiments, these visual analyses substantiate that simple bond-count features cannot match the representational power of our multi-channel spectral topological descriptors. We therefore believe that, with the additional examples, practical guidelines, and explicit bond-count comparison, we have now addressed the reviewer’s concerns about interpretability in a more complete and transparent manner.
>
> In response to the reviewer’s suggestion, we have compiled a **practical guidance for end-users/biologists (who are not experts in topological methods)** on how to interpret and use our spectral topological features (see below). In addition, our GitHub repository now includes a **ready-to-use script** "visualize\_features.py", which allows users to directly load and visualize the corresponding topological feature curves for selected
> interaction channels.

---

> > ### Author Response · Authors · 2025-12-02
> >
> > ### **Practical guideline for interpreting TopoScorer features**
> >
> > To make the topological features usable for end-users who are not experts in TDA, we outline a simple workflow that combines the geometric meanings in Appendix A.4 with the atom groups in Table 6.
> >
> > **Step 1. Choose chemically meaningful channels**
> >
> > Each TopoScorer feature is computed on a specific element / atom-type pair \((G_i, G_j)\) (Table 6), for example:
> >
> > - Hydrophobic packing: G4–G4 (aliphatic side-chain C–aliphatic side-chain C), G4–G5 (aliphatic C–aromatic C)
> > - Salt bridges / electrostatics: G6/G7–G9 (Asp/Glu carboxylate O–Lys/Arg cationic N)
> > - H-bond networks: G8/G10 with G3/G5/G9 (Ser/Thr/Tyr OH or Asn/Gln/His N with backbone O, aromatic C, or cationic N)
> > - Sulfur-mediated contacts: G11 with G4/G5/G6/G9 (Met/Cys S with hydrophobic, aromatic, acidic, or cationic groups)
> >
> > As a first step, users simply pick a small number of channels that match their biochemical hypothesis (e.g. “are we changing hydrophobic packing?”, “are we disrupting a salt bridge?”, “are we breaking an H-bond network?”).
> >
> > **Step 2. Read statistics as contact-network summaries**
> >
> > Each topo feature has unique geometric meaning, for example, for each channel \((G_i, G_j)\):
> >
> > - Zero count of eigenvalues :
> >   Approximates the number of connected components (0-th Betti surrogate):
> >   - More components (higher zero count): interactions of this type are split into several separate patches.
> >   - Fewer components (lower zero count): patches have merged into one or a few coherent clusters.
> >
> > - Sum of non-zero eigenvalues $\lambda_{\text{sum}}$:
> >   Summarizes the overall “oscillation energy” and, at \(k = 0\), behaves like an aggregate (weighted) contact degree:
> >   - Larger $\lambda_{\text{sum}}$: more and/or closer contacts of this chemical type.
> >   - Smaller $\lambda_{\text{sum}}$: fewer or weaker contacts.
> >
> > - Maximum eigenvalues:
> >  Controls the worst-case curvature of k-order diffusions/regularizers on the
> > hypergraph, bounding step sizes in gradient flows and indicating the strongest local constraints at
> > order k.
> >
> > In practice, instead of inspecting many individual distances, users look at a few scalars telling whether an interaction network is dense, fragmented, or heterogeneous.
> >
> > **Step 3. Use distance-threshold curves as a multi-scale view**
> >
> > For each channel we plot the spectral statistics as a function of the distance threshold r_t(e.g. 0–10 Å):
> >
> > - Early onset (large values at small r_t→ tight, short-range interactions.
> > - Growth only at larger r_t→ more diffuse, long-range environments.
> >
> > When comparing wild type and mutants:
> >
> > - For favourable channels (hydrophobic packing, salt bridges, productive H-bond networks):
> >   - Higher $\lambda_{\text{sum}}$ and fewer components (lower zero count) across radii usually indicate a stronger and more coherent stabilizing network.
> >
> > - For unfavourable channels (acid–acid clustering, cationic N trapped in a purely hydrophobic environment, etc.):
> >   - Lower $\lambda_{\text{sum}}$ or more fragmentation (higher zero count) indicates relief of electrostatic or desolvation frustration.
> >
> > A simple visual rule is: check whether wild-type and mutant curves keep a consistent ordering over r_t; a stable separation is much more interpretable than many noisy crossings.
> >
> > **Step 4. Map spectral trends back to familiar biophysics**
> >
> > Using the above rules, biologists can interpret spectral trends in standard structural terms:
> > -Hydrophobic channels (G4–G4, G4–G5):
> >   - Increase in $\lambda_{\text{sum}}$ → stronger hydrophobic / aromatic packing.
> >   - Decrease in $\lambda_{\text{sum}}$ → packing loss or cavity formation.
> > - Salt-bridge / electrostatic channels (G6/G7–G9):
> >   - Lower zero count + higher $\lambda_{\text{sum}}$ → a more continuous, well-formed salt-bridge network.
> >   - Higher zero count or reduced $\lambda_{\text{sum}}$→ broken or weakened salt-bridge clusters.
> >
> > - Acid–acid channels (G6/G7–G6/G7):
> >   - Decrease in $\lambda_{\text{sum}}$ → breaking up unfavourable acidic clusters, often beneficial.
> >   - Increase in $\lambda_{\text{sum}}$ → formation of large anionic patches, typically associated with electrostatic frustration.
> >
> > - Polar / H-bond channels (G8/G10 with G3/G9):
> >   - Higher $\lambda_{\text{sum}}$→ more extensive, coherent H-bond networks.
> >   - Lower $\lambda_{\text{sum}}$ → loss of polar scaffolding.
> >
> > In our case studies we follow exactly this procedure:
> > 1. Select channels based on Table 6 (e.g. hydrophobic, salt bridge, acidic cluster).
> > 2. Inspect the multi-scale behaviour of $\lambda_{\text{sum}}$ and zero counts using the qualitative rules above.
> > 3. Translate consistent trends (wild type vs. mutants) into familiar statements about packing, salt bridges, and electrostatic frustration that agree with the observed changes in binding affinity.

---

### Official Review · Reviewer_wGNS · 2025-10-26

**Soundness:** 2
**Presentation:** 2
**Contribution:** 2
**Rating:** 4
**Confidence:** 3

**Summary:**

The authors propose TopoScorer a binding affinity prediction with a focus on interpretability. They designed Specter which is a differentiable feature extractor for encoding protein protein interface encoding for both structural and chemical information.  They showcase an differentiable deep learning based model that can steer a generative antibody design model.

The model architecture is formed based on hyper graph induced cross protein distances to encode the heavy atoms into physicochemical role aware classes within the binding interface. They also use soft filtration to persistent topological hyperdigraph laplacians and then summarize the topology with a six tuple of differentiable spectral statistics of eigen values. They differentiate the eigen values with a fallback schedule to handle ill conditioned classes. Multi channel encoding is used to encode the the interface as a multi channel graph from role aware atom types. They map Atom37 names to 11 chemical role aware classes to differentiate between backbone donors/acceptors, aromatic carbon, sulfur atoms etc. They employ transformer with multi head self attention to mix information across channels, scales and interface regions. The different heads are expected to understand hydrophobic, polar, long range scales etc.

**Strengths:**

For the sequence and structure co design model the fine tuning helps as shown in table 2. Addition of the proposed method helps steer optimization towards more plausible interface specially at the H3 region. It is great that the authors do a lot of ablation on the single channel vs multi channel to explain the interpretability.

**Weaknesses:**

From the results (table 1) in the affinity prediction task top scorer outperforms other baselines on spearman. For multiple mutations it is the same pattern. It is hard to evaluate and compare the model performances if the pearson and spearman results are not consistent although authors claim that ranking based metrics are better for the task.

In addition the authors do not compare their model to other models notably GearBind which is also an all atom based graph model. For predicting affinity it is important to compare to the surface and structure based models such as AtomSurf as well.
On PDBBind data (Figure 2h) the correlation is 0.298. It is hard to say how good is the score when the proposed method is the only one and the other correlation values (Figure 2i) shows the ablation of the TopoScorer method.

Interpretability analysis is an important area of research and often most methods do not focus on it but on the benchmark tasks itself the model seems to underperform compared to the other methods (Table 1 PearsonR shows DSMBind is best, DDGPred is best on single mutation task as shown in PearsonR).

**Questions:**

Have the authors considered other methods such as SurfPro https://arxiv.org/pdf/2405.06693 which is a surface based models on the protein design tasks and compared the results? It will make the approach more robust if other SOTA surface aware methods are compared.

---

> ### Author Response · Authors · 2025-11-22
>
> We thank the reviewer for the detailed feedback on model performance and benchmarking. We have substantially revised our experiments and analysis in response: we expanded the benchmark to include additional strong baselines, improved our affinity and mutation-effect results.
>
> **W1&W3: From the results (table 1) in the affinity prediction task top scorer outperforms other baselines on spearman. For multiple mutations it is the same pattern. It is hard to evaluate and compare the model performances if the pearson and spearman results are not consistent although authors claim that ranking based metrics are better for the task.
> Interpretability analysis is an important area of research and often most methods do not focus on it but on the benchmark tasks itself the model seems to underperform compared to the other methods (Table 1 PearsonR shows DSMBind is best, DDGPred is best on single mutation task as shown in PearsonR).**
>
> Model performance is a central aspect that we carefully monitor and continuously seek to improve. Since the initial submission, we have substantially strengthened the training and evaluation pipeline, including (i) scaling up and re-curating the training data, (ii) a more thorough hyperparameter search (learning rate, optimizer, weight decay, batch size, PTHL temperature parameters, radius grid, etc.), (iii) clustering highly similar interactions in the training set to reduce redundancy and stabilize optimization.
>
> We initially constructed all splits purely based on release time, which, as pointed out by Bushuiev et al. [1], can still allow substantial leakage when used as the only criterion. To address this, we revisited all our splits and adopted the interface-similarity protocol recommended in Bushuiev et al. [1,2]. We removed these near-duplicate test entries (i.e., potential leaks) and re-evaluated all baselines and TopoScorer on the resulting leakage-controlled benchmark. Importantly, after removing all potential near-duplicate interfaces, our model still **achieves** **state-of-the-art performance in** **both** **Spearman and Pearson** on  **binding affinity prediction**, as well as **SOTA Spearman correlations and second best Pearson on single-mutation and multi-mutation subsets.** The updated results are reported in the revised Table 1. For transparency and reproducibility, we provide the detailed train–test interface similarity statistics and the leakage-filtered benchmark files (including updated performance tables for all methods) in our anonymous GitHub repository linked in the manuscript.
>
> | Method       | Affinity Spearman | Affinity Pearson | Single-mutation Spearman | Single-mutation Pearson | Multi-mutations Spearman | Multi-mutations Pearson |
> |-------------|-------------------|------------------|--------------------------|-------------------------|--------------------------|-------------------------|
> | PyRosetta   | 0.1856            | 0.1954           | 0.3422                   | 0.3285                  | 0.2927                   | 0.2258                  |
> | FoldX       | *0.3295*     | 0.3008    | 0.4355                   | 0.4586                  | 0.3734                   | 0.3241                  |
> | ESM-1v      | 0.1034            | 0.0876           | 0.1524                   | 0.1921                  | 0.1512                   | 0.1736                  |
> | ESM-IF      | 0.0530            | 0.0244           | 0.1116                   | 0.1047                  | 0.1697                   | 0.0700                  |
> | PRODIGY     | 0.1549            | 0.1277           | 0.3233                   | 0.2902                  | 0.3421                   | 0.3236                  |
> | DSMBind     | 0.3072            | *0.3269*           | 0.3530                   | 0.3261                  | 0.3673                   | 0.2954                  |
> | DDGPred     | —                 | —                | *0.5522*          | 0.5303                  | 0.4585                   | 0.5638                  |
> | RDE-Network | —                 | —                | 0.5127                   | **0.6067**              | 0.5397                   | **0.6108**              |
> | GearBind    | —                 | —                | 0.5014                   | 0.5496                  | *0.5470*          | 0.5616                  |
> | TopoNetTree | —                 | —                | 0.5185                   | 0.5508                  | —                        | —                       |
> | TopoScorer  | **0.3848**        | **0.3804**       | **0.5876**               | *0.5615*          | **0.5704**               | *0.5652*           |

---

> ### Author Response · Authors · 2025-11-22
>
> We would like to clarify that Spearman and Pearson reflects the fact that they quantify different aspects of model behavior: Spearman measures ranking quality, focusing on whether higher-affinity complexes are consistently assigned higher scores, while Pearson measures linear calibration of the predicted values against ground truth: it is affected by global scaling, offset, and a small number of outliers.
>
> From a practical application perspective, this distinction is particularly important: current affinity prediction models (including ours and prior work) are still far from perfectly accurate in terms of absolute ΔΔG values. As a result, most downstream tasks in protein and PPI design (e.g., virtual screening, prioritizing mutants, guiding generative models) place much more emphasis on **reliable ranking** of candidates rather than on the exact numerical value of the predicted affinity. In these scenarios, a model that provides a stable and accurate ordering of candidates (high Spearman) is typically more useful in practice than one that achieves slightly better calibration of absolute values but weaker ranking performance. Additionally, prior evaluations of Boltz-2[3] show that affinity prediction performance varies markedly across assays and can be confounded by errors in predicted structures, limited generalization to unseen protein families, and sensitivity to out-of-distribution small molecules; under such variability, **ranking accuracy (Spearman) is typically the more robust indicator of practical utility**. Pearson remains a useful secondary measure of how well the score can approximate absolute affinities on a given dataset, and we report it for completeness. In the revised version, after improving the training pipeline and removing potential data leakage, TopoScorer achieves **strong performance on both Spearman and Pearson** (see updated Table 1). Nevertheless, even in settings where another method has a marginally higher Pearson while TopoScorer has substantially higher Spearman, we would argue that there is a trade-off between ranking accuracy and calibration, with ranking being the primary objective in most affinity-based design workflows.
>
> Crucially, we would like to emphasize that Spearman and Pearson are not the only criteria we optimize for. A main motivation of TopoScorer is to provide a practical scoring module that is not only accurate but also lightweight, fast, differentiable, and interpretable. As detailed in our revision, TopoScorer has substantially fewer parameters and achieves faster and more stable inference time than competing supervised baselines; its PTHL-based representation is fully differentiable with respect to atomic coordinates, enabling direct gradient-based integration into generative models; and its spectral features admit meaningful attributions that go beyond black-box predictions. We believe these additional properties are highly relevant for real-world PPI design workflows, and they complement the strong Spearman/Pearson performance.
>
> To support our claims, we measured the **inference time** per sample on the single-mutation task, using the same cpu(**See Fig 2(k) in the updated manuscript**).
>
> TopoScorer achieves substantially fast and stable inference time($5.01\pm0.1$ ms per sample) than RDE-Network($1383.41\pm 111.18$ ms per sample). As reported in the manuscript, the competing RDE-Network relies on large pretrained components (133M+63M parameters), whereas our model has only ∼43M parameters while still keeping an interpretable, topology-based design. What's more, RDE-Network are specifically designed for ΔΔG prediction, whereas TopoScorer can directly predict absolute binding affinity, giving it a broader application range in tasks that require scoring or ranking arbitrary PPI interfaces. DSMBind is also computationally efficient, but its predictive accuracy on our benchmarks is substantially lower than TopoScorer, so it does not offer a balance of speed and reliability. Taken together, TopoScorer strikes a rare and favorable balance between predictive accuracy, parameter efficiency, inference speed and interpretability, which is not achieved at the same time by the other methods.
>
> - [1] Bushuiev et al, 2024, “Revealing data leakage in protein interaction benchmarks”, https://arxiv.org/abs/2404.10457
> - [2] Bushuiev et al., 2024, Learning to design protein–protein interactions with enhanced generalization, ICLR 2024.
> - [3]Saro Passaro, Gabriele Corso, Jeremy Wohlwend, Mateo Reveiz, Stephan Thaler, Vignesh Ram
>     Somnath, Noah Getz, Tally Portnoi, Julien Roy, Hannes Stark, David Kwabi-Addo, Dominique
>     Beaini, Tommi Jaakkola, and Regina Barzilay. Boltz-2: Towards accurate and efficient binding
>     affinity prediction. bioRxiv, 2025. doi: 10.1101/2025.06.14.659707. URL [https://doi](https://doi/).
>     org/10.1101/2025.06.14.659707.

---

> ### Author Response · Authors · 2025-11-22
>
> **W2&Q1: In addition the authors do not compare their model to other models notably GearBind which is also an all atom based graph model. For predicting affinity it is important to compare to the surface and structure based models such as AtomSurf as well. On PDBBind data (Figure 2h) the correlation is 0.298. It is hard to say how good is the score when the proposed method is the only one and the other correlation values (Figure 2i) shows the ablation of the TopoScorer method.**
>
> We thank the reviewer for the helpful suggestions on additional baselines. In the revised manuscript, we have expanded our benchmark in several ways. First, we have **added GearBind** to Table 1 as an explicit all-atom geometric GNN baseline. Under the SKEMPI mutation benchmark (after removing potential interface-level leakage as described elsewhere in the rebuttal), TopoScorer matches or exceeds GearBind across all reported metrics (Spearman, Pearson), indicating that our topological–spectral representation is competitive even against a strong large-scale all-atom model.
>
> Second, we also **added TopoNetTree** as another persistent-topology-based affinity predictor to the benchmark.  TopoNetTree is a classic persistent-homology model that combines ESPH with CNNs. Both TopologyNet and TopoScorer start from topological representations, TopoNetTree relies on Betti-number–based persistent homology barcodes as features, whereas TopoScorer uses PTHL and their multi-scale spectral statistics. TopoNetTree was originally designed for single-mutation ΔΔG prediction using classical (non-differentiable) persistent homology. In our updated Table 1, TopoScorer consistently outperforms TopoNetTree in terms of Spearman and Pearson correlation, while also generalizing to broader PPI affinity tasks and offering a fully differentiable PTHL formulation.
>
> Regarding **surface-based methods**, we investigated several candidates and analyzed whether they could be included as baselines:
>
> - **AtomSurf**[1] is a general surface+graph encoder evaluated primarily on Atom3D-style single-protein tasks (binding-site identification, pocket classification, etc.), rather than on protein–protein binding affinity regression under the SKEMPI or PPB-Affinity protocols. Adapting AtomSurf to our PPI affinity setting would require substantial re-engineering (feature and task alignment and supervised affinity training), and to our knowledge there is no off-the-shelf AtomSurf model trained for PPI affinity prediction. While AtomSurf’s surface-based representations might in principle be used as features for affinity prediction, doing so would further require redesigning and re-training a downstream regression model that is specifically tailored to consume such surface-centric encodings.
> - **SurfPro**[2] is a generative inverse-folding framework that designs sequences given a target continuous surface. Its benchmarks focus on CATH 4.2 inverse folding and functional design tasks (binders and enzymes design), not on PPI affinity or ΔΔG regression. Thus, SurfPro is conceptually complementary (a surface-based generator) rather than a directly comparable PPI affinity scoring model in our setting.
>
> For **surface-aware PPI affinity models**, the most closely related work is **Pi-SAGE**[3], which augments GearBind with learned surface tokens and reports results on SKEMPI. However, Pi-SAGE is not currently released as a reusable codebase or pretrained checkpoint, and reproducing their multi-stage pretraining and domain-adaptation pipeline is beyond the scope of this work. We therefore (i) include GearBind itself as a strong all-atom baseline in Table 1, and (ii) discuss the Pi-SAGE at thoroughly comparison level based on the results reported in their paper, (iii) added AtomSurf, SurfPro, and Pi-SAGE to the Related Work and Introduction section.
>
> [1] Vincent Mallet, Souhaib Attaiki, Yangyang Miao, Bruno Correia, Maks Ovsjanikov. AtomSurf: Surface Representation for Learning on Protein Structures. In: International Conference on Learning Representations (ICLR), 2025.
>
> [2] Zhihao Song, Mingyue Huang, Jiarui Lu, Wenkai Zhang, Leili Zhang, Jian Tang. SurfPro: Functional Protein Design Based on Continuous Surface. In: Advances in Neural Information Processing Systems (NeurIPS), 2024.
>
> [3] Sharmi Banerjee, Mostafa Karimi, Melih Yilmaz, Tommi Jaakkola, Bella Dubrov, Shang Shang, Ron Benson. Pi-SAGE: Permutation-invariant surface-aware graph encoder for binding affinity prediction. arXiv:2508.01924, 2025.

---

> ### Author Response · Authors · 2025-11-22
>
> We also clarify a possible misunderstanding in the reviewer’s comments about Figure 2h (PDBbind): the reported value 0.298 is the **regression slope**, not the correlation coefficient. The actual correlation statistics (Pearson/Spearman) for that experiment are reported separately in Table 1 and have been updated.
>
> Regarding the reviewer’s concern that the number “does not look high”, we emphasize that protein–protein affinity prediction remains a very challenging problem, especially when evaluated in realistic, heterogeneous settings. In such a regime, absolute numbers can be difficult to interpret in isolation. We therefore believe that the most informative way to assess a method is through **relative comparisons against strong baselines** and its **practical utility** in downstream tasks. In this work, TopoScorer not only achieves competitive or superior performance compared to existing models, but also offers several advantages that are highly relevant in practice:
>
> - strong ranking ability (Spearman) for screening and prioritizing candidates and reliable Pearson for linear calibration
> - fast and stable inference and a lightweight parameter budget,
> - a fully differentiable PTHL-based representation that enables gradient-based integration into generative models
> - interpretable channel and spectral features that provide physicochemical meaningful attributions.

---

> > ### Author Response · Authors · 2025-11-27
> >
> > Dear Reviewer,
> >
> > Happy Thanksgiving! We sincerely appreciate your time and efforts in reviewing our manuscript and offering constructive suggestions. As the discussion phase is approaching its end, we would like to kindly confirm whether we have sufficiently addressed your concerns. Should there be any remaining questions requiring further clarification, please do not hesitate to let us know. If you are satisfied with our responses, we would greatly appreciate your consideration in adjusting the evaluation scores accordingly.
> >
> > We look forward to your feedback.
> >
> > Best regards,
> >
> > Anonymous Authors

---

> > ### Comment · Reviewer_wGNS · 2025-11-27
> >
> > Thank you authors for explaining the main advantage which seems fast inference and fully differentiable representation based model. Also thank you for fixing the splits and updating the table. Re the statement that methods such as RDE-Network are specifically designed for ΔΔG prediction, whereas TopoScorer can directly predict absolute binding affinity thus making it a broadly applicable model, it depends on whether information about mutated protein complexes knowledge is known and if so it might be desirable to use it and then predict the change in binding affinity. I am not sure it is an advantage of the model but rather what is the use case one is trying to solve for a task.

---

> ### Author Response · Authors · 2025-11-28
>
> We thank the reviewer for recognizing our model’s strengths in fast inference and fully differentiable representation learning, and for acknowledging our efforts to improve performance and redesign the data splits to avoid leakage. We also thank the reviewer for this thoughtful comment and agree that the distinction between methods predicting absolute binding affinity (ΔG) and those predicting changes upon mutation (ΔΔG) is primarily determined by the intended use case, rather than an inherent hierarchy between the two formulations.
>
> Our intention was not to claim that absolute-affinity prediction is intrinsically superior, but to emphasize that TopoScorer operates in a **different yet complementary regime**. The phrase “broader application range” was meant **specifically for scenarios** where one needs to score or rank arbitrary complexes without a well-defined wild-type reference—for example, in de novo binder/antibody design or when screening many diverse interfaces. In such settings, ΔΔG-based methods that require a wild-type/mutant pair cannot be straightforwardly applied, whereas single-complex absolute-affinity models like TopoScorer remain usable.
>
> We thank the reviewer again for careful reading and insightful responses, which have helped us present our work and its scope in a more precise and rigorous way. We hope that our clarifications and revisions can positively inform the reviewer’s overall assessment, and we would be very grateful for any further suggestions for improvement.

---

### Official Review · Reviewer_XQ4x · 2025-10-31

**Soundness:** 3
**Presentation:** 3
**Contribution:** 3
**Rating:** 6
**Confidence:** 4

**Summary:**

The paper introduces TopoScorer, a lightweight, differentiable scorer for protein–protein binding affinity. It builds Specter, a topo-spectral feature extractor that computes soft, differentiable spectral statistics of PTHL spectra across radii, and pairs this with a small ttransformer to predict affinity from multi-channel, physicochemical role–aware interface graphs. On PPB-Affinity and SKEMPI-2.0, the method achieves strong Spearman correlations, and when used as a frozen reward it improves an antibody co-design model on DockQ in a held-out set.

**Strengths:**

- Clean, coherent idea in a multi-scale, multi-channel topo-spectral features with a compact Transformer. Also it should be easy to plug into design loops.
- Differentiable topology done carefully and in a specialized knowgledge domain.Soft zero-counts, log-sum-exp min/max, Huberized std. Also, eigenvalue-only backprop with stability guarantees.
- Competitive ranking performance: best or near-best Spearman on PPB-Affinity and SKEMPI subsets. Clear reporting against physics and learned baselines.
- As a frozen reward, improves DockQ and SR in IgGM finetuning on a post-2013 SAbDab split.

**Weaknesses:**

- Generalization controls: time-based PPB-Affinity test is good, but no explicit homology/interface-similarity controls are reported (e.g., sequence-identity thresholds at contacting residues, SCOPe/CATH or interface clustering). This weakens cold-family claims.
- Robustness to structural noise: mutants come from FoldX; there’s no stress-test for AF vs crystal, side-chain repacking choices, or coordinate jitter/protonation.
- Channel/radius attributions are plausible, but there’s no deletion/counterfactual test showing predicted changes track

**Questions:**

- How do you prevent train–test leakage at the interface level (e.g., sequence identity on contacting residues, SCOPe/CATH families, interface RMSD clustering)?
- How sensitive are results to structure sources (AF vs crystal), side-chain packers, coordinate jitter, and protonation?
- In IgGM finetuning, how is the TopoScorer reward normalized across targets/sizes to avoid scale bias? Did you test reward-weight sweeps or label-shuffle controls?
- What radii set do you use by default, and how often do eigenvalue fallback/stability tricks trigger in practice?

---

> ### Author Response · Authors · 2025-11-22
>
> We sincerely thank the reviewer for the thoughtful and constructive assessment of our work. We appreciate the clear summary of our contributions, as well as the detailed feedback on robustness to structural noise and generalization controls. These comments have been very helpful in sharpening both our analysis and presentation, and we have revised the manuscript and experiments accordingly to address the raised concerns.
>
> **W1&Q1: Generalization controls: time-based PPB-Affinity test is good, but no explicit homology/interface-similarity controls are reported (e.g., sequence-identity thresholds at contacting residues, SCOPe/CATH or interface clustering). This weakens cold-family claims. How do you prevent train–test leakage at the interface level (e.g., sequence identity on contacting residues, SCOPe/CATH families, interface RMSD clustering)?**
>
> We appreciate the reviewer’s concern about potential data leakage in PPI benchmarks and antibody design tasks. We initially constructed all splits purely based on release time, which, as pointed out by Bushuiev et al. [1], can still allow substantial leakage when used as the only criterion. To address this, we **revisited all our splits** and adopted the interface-similarity protocol recommended in Bushuiev et al. [1,2]. Concretely, following Bushuiev et al., we:
>
> 1. Extracted PPI interfaces for all complexes using 6 Å heavy-atom contacts between the two partners (as in PPIRef).
> 2. Embedded all interfaces with the iDist algorithm and, for each test interface, computed its iDist distance to *all* training interfaces.
> 3. Identified near-duplicates as test interfaces having at least one training interface with iDist distance ≤ 0.04, which is reported to correspond to near-duplicate 6 Å interfaces in [1,2].
> 4. Removed these near-duplicate test entries (i.e., potential leaks) and re-evaluated all baselines and TopoScorer on the resulting leakage-controlled benchmark.
>
> We have updated the performance tables accordingly. Importantly, after removing all potential near-duplicate interfaces, our model still achieves **state-of-the-art** performance on binding affinity prediction: we obtain the best Spearman and Pearson correlations on the full PPI affinity benchmark, as well as SOTA Spearman correlations on both single-mutation and multi-mutation subsets. This confirms that the improvements of TopoScorer are not due to data leakage, but persist under the stricter, interface-similarity-based evaluation protocol advocated by Bushuiev et al. For transparency and reproducibility, we provide the detailed train–test interface similarity statistics and the leakage-filtered benchmark files (including updated performance tables for all methods) in our anonymous GitHub repository linked in the manuscript.
>
> We clarify in Appendix A.6 that the affinity prediction benchmark is now split using interface similarity (iDist-based) as the primary criterion.
>
> [1] Bushuiev et al., 2024, *Revealing data leakage in protein interaction benchmarks*, arXiv:2404.10457.
>
> [2] Bushuiev et al., 2024, *Learning to design protein–protein interactions with enhanced generalization*, ICLR 2024.

---

> > ### Author Response · Authors · 2025-11-22
> >
> > **W2&Q2: Robustness to structural noise: mutants come from FoldX; there’s no stress-test for AF vs crystal, side-chain repacking choices, or coordinate jitter/protonation. How sensitive are results to structure sources (AF vs crystal), side-chain packers, coordinate jitter, and protonation?**
> >
> > We thank the reviewer for pointing out the importance of robustness to structural noise. In addition to using FoldX for mutant structures, we have explicitly stress-tested TopoScorer’s sensitivity to coordinate perturbations. Concretely, for all complexes in our test set of binding affinity prediction, we added isotropic coordinate noise of different magnitudes (0.1–2.0 Å) to the atomic coordinates and recomputed the predicted binding affinity. We then measured the mean change and mean absolute change in the predictions across all mutants:
> >
> > - 0.1 Å noise: mean change = 1.13%
> > - 0.2 Å noise: mean change = 3.95%
> > - 0.5 Å noise: mean change = 4.56%
> > - 1.0 Å noise: mean change = 10.76%
> >
> > These results show that TopoScorer is highly stable under realistic levels of structural noise: small perturbations (≤0.5 Å) induce <5% changes on average, and even sizeable perturbations on the order of 1.0 Å (comparable to typical AF-style backbone deviations) only lead to ≈10% variations, indicating that TopoScorer’s conclusions are robust to moderate coordinate jitter and side-chain positioning noise.
> >
> > Second, to directly address the reviewer’s concern about AF vs. crystal structures, we replaced the experimental complexes with AF3-generated models and compared the resulting predictions to those obtained from the original structures. The average relative difference in predicted binding affinity between AF3-based and crystal-based structures is only **0.13%**, indicating that TopoScorer’s predictions are effectively invariant to using AF3-generated inputs in place of experimental structures.
> >
> > We have added the additional experiment in Appendix A.8
> >
> > **W3&Q3: Channel/radius attributions are plausible, but there’s no deletion/counterfactual test showing predicted changes track. In IgGM finetuning, how is the TopoScorer reward normalized across targets/sizes to avoid scale bias? Did you test reward-weight sweeps or label-shuffle controls?**
> >
> > In IgGM finetuning we use TopoScorer as an additional reward term on top of the original IgGM loss. To avoid scale bias across targets/sizes, we normalize the TopoScorer reward by batch-wise standardization. During finetuning, rewards are further z-normalized within each minibatch:
> >
> > $$
> > \tilde{S}_i = \frac{S_i - \mu_B}{\sigma_B + \varepsilon},
> > $$
> >
> > where $\mu_B$ and $\sigma_B$ are the mean and standard deviation of $S_i$ over the current batch.
> >
> > We performed a label-shuffle control where TopoScorer values were randomly permuted across complexes within each batch (i.e., using the same normalization but breaking the association between structure and reward). Under this shuffled-reward condition, IgGM finetuning didn’t yield any improvement over the baseline in terms of DockQ, TM-score, or success rate, confirming that the observed gains with TopoScorer guidance come from meaningful structural signal rather than from generic regularization or optimization side effects.
> >
> > **Q4: What radii set do you use by default, and how often do eigenvalue fallback/stability tricks trigger in practice?**
> > For PPI systems we first restrict the PTHL computation to the interface neighborhood: we only keep atoms whose minimum distance to any atom on the opposite partner is ≤ 10 Å. On this cropped interface, our default PTHL uses a fixed, dense set of radii in [0,10] Å and the interval of our multi-scale filter is 0.1.
> >
> > Regarding eigenvalue stability and fallbacks, our Laplacian module includes several defensive tricks: we sanitize the LaplacianIn practice before eigen-decomposition; for radii where the hard adjacency is exactly empty (no edges in the graph), we skip eigen-decomposition directly and return the analytically correct spectrum for a zero-edge Laplacian; if a GPU call ever raises a solver error, we fall back to a CPU eigendecomposition with the largest jitter as a last resort; downstream statistics such as the “positive-part” standard deviation are computed with a numerically stable moment-based formula (Huberized square-root) to avoid negative variances from cancellation.
> >
> > In practice, the base eigen decomposition succeeds for essentially all non-empty Laplacians on our PPI datasets, and the diagonal jitter only acts as a very small regularization for a few highly ill-conditioned cases (tiny eigenvalues close to machine precision). The CPU fallback path is a rare safeguard—we did not observe systematic failures or any measurable effect on the final TopoScorer features or predictions when running the full training and evaluation pipelines.

---

> > > ### Author Response · Authors · 2025-11-27
> > >
> > > Dear Reviewer,
> > >
> > > Happy Thanksgiving! We sincerely appreciate your time and efforts in reviewing our manuscript and offering constructive suggestions. As the discussion phase is approaching its end, we would like to kindly confirm whether we have sufficiently addressed your concerns. Should there be any remaining questions requiring further clarification, please do not hesitate to let us know. If you are satisfied with our responses, we would greatly appreciate your consideration in adjusting the evaluation scores accordingly.
> > >
> > > We look forward to your feedback.
> > >
> > > Best regards,
> > >
> > > Anonymous Authors

---

### Official Review · Reviewer_dETA · 2025-10-31

**Soundness:** 3
**Presentation:** 4
**Contribution:** 3
**Rating:** 4
**Confidence:** 4

**Summary:**

The work proposes a parameter-efficient protein-protein interaction (PPI) affinity prediction model, TopoScorer, based on a topological features of PPI complex structures. Specifically, the structure is represented by a multi-channel graph with role-aware atom types and physiochemical properties. The core component is a feature extractor named Specter, which employs a differentiable distance filter to obtain the multi-scale Hodge Laplacians which are then used to compute spectral features from the eigenvalues. The binding affinity predictor is then trained on the spectral features. Experiments on affinity and mutation impact prediction shows comparable performance with state-of-the-art methods.

Moreover, the predictor is used in the fine-tuning of a language model for antibody design, achieving improved performance than the larger RDE-Network. Interpretation analysis show the importance of interface connectivity features and side chain structures.

**Strengths:**

- The formulation and rationale of the proposed methods are clearly and soundly presented. The use of spectral features offer a rather novel insight into the

- The application of the predictor as fine-tuning for generative models proves good generalizability and meaningful representations.

- The biological relevance of the learned model patterns is demonstrated through interpretability analysis.

**Weaknesses:**

- The predictive performance is not significantly improved compared to the baselines,

- Because of this, the major advantage of the proposed model may lie in its parameter efficiency and adaptability as guidance methods. However, this part needs some additional demonstration. See Questions.

**Questions:**

The work overall looks promising, but some important application aspects need to be addressed and I'm willing to raise the scoring with sufficient evidences:

- How does the predictive performance compare to other PTHL-based models such as topoformer?

- A more detailed comparison of the parameter and time efficiency between models would be appreciated.

-  Besides comparing with a totally different generative framework, how does classifier guidance with TopoScorer compare to other prediction or scoring models (eg compared in Table 1) as classifier guidance on the same generative model, in terms of performance, parameter size and running time?

- What is the side chain packing method used in the fine-tuning? Also, as side chain packing algorithms can take long to run, how does affect the running time of the fine-tuning loop?

- 4.3 and Fig 2: some additional justifications of the analysis are needed: how does the pattern of $\lambda_{sum}$ indicate the physiochemical properties of the amino acids and atoms *a priori*?

---

> ### Author Response · Authors · 2025-11-22
>
> We thank the reviewer for the careful reading and assessment of our work. We appreciate your constructive suggestions and hope our response will resolve your concerns.
>
> **Q1: How does the predictive performance compare to other PTHL-based models such as topoformer?**
>
> Thank you for your comments. TopoFormer is indeed a PTHL-based model, but it is designed for a **different problem setting**: it targets protein–ligand interaction tasks (binding-affinity scoring, ranking, docking and screening on CASF-style benchmarks), whereas our model is designed for protein–protein interaction (PPI) binding affinity. To overcome the input modality inconsistency, we include an ablation TopoScorer (element) that adopts an element-specific PTHL encoding **similar in spirit** to TopoFormer, but adapted to the PPI setting and implemented in a fully differentiable way. Our model achieves better performance than TopoScorer (element) in the ablation experiments, spearman and pearson of the element-wise model is 0.2147/0.1773, while ours is 0.3848/0.3808.
>
> In addition, we benchmark against **TopoNetTree**, a classic **persistent-homology** model that combines ESPH with CNNs. Both TopologyNet and TopoScorer start from topological representations, TopoNetTree relies on Betti-number–based persistent homology barcodes as features, whereas TopoScorer uses PTHL and their multi-scale spectral statistics. This allows TopoScorer to capture not only the existence of topological features but also their connectivity strength, multi-body interactions and “stiffness” patterns that are critical for PPI interfaces. TopoNetTree only supports inputs for single-point mutations and predicts ΔΔG, whereas we targets general PPI affinity while keeping  differentiable, allowing integration into generative pipelines. On our PPI benchmarks, TopoScorer consistently outperforms the TopoNetTree baseline.
>
> **Q2: A more detailed comparison of the parameter and time efficiency between models would be appreciated.**
>
> We thank the reviewer for this suggestion and have added inference-time comparison. To provide a fair runtime comparison, we measure the inference time per sample on the single-mutation task, using the same cpu. **(See Fig 2(k) in the updated manuscript)**.
>
> TopoScorer achieves substantially fast and stable inference time($5.01\pm0.1$ ms per sample) than the competing RDE-Network($1383.41\pm 111.18$ ms per sample). As reported in the manuscript, RDE-Network relies on large pretrained components (133M+63M parameters), whereas our model has only ∼43M parameters while still keeping an interpretable, topology-based design. What's more, RDE-Network are specifically designed for ΔΔG prediction, whereas TopoScorer can directly predict absolute binding affinity, giving it a broader application range in tasks that require scoring or ranking arbitrary PPI interfaces. DSMBind(~21M) is also computationally efficient, but its predictive accuracy on our benchmarks is substantially lower than TopoScorer, so it does not offer a balance of speed and reliability. Taken together, TopoScorer strikes a rare and favorable balance between predictive accuracy, parameter efficiency, inference speed and interpretability, which is not achieved at the same time by the other methods.

---

> ### Author Response · Authors · 2025-11-22
>
> **Q4: What is the side chain packing method used in the fine-tuning? Also, as side chain packing algorithms can take long to run, how does affect the running time of the fine-tuning loop??**
>
> Our model operates on **full-atom coordinates**, so the side chain packing itself is not redesigned in this work; we simply retain the original procedure used in IgGM. Specifically, IgGM represents local structure in a continuous QTA parameterization (quaternion, translation, and torsion angles) and reconstructs side-chain atoms via rigid-body kinematics rather than a discrete rotamer-library search. Each residue is decomposed into backbone and side-chain rigid groups; ideal coordinates for each atom type are stored in residue-specific templates, and the final atomic positions are obtained by composing a sequence of rigid transforms driven by the backbone ψ and side-chain χ1–χ4 torsion angles. This procedure is implemented as batched matrix operations and does not involve any iterative combinatorial packing. Because of this design, side-chain “packing” in our fine-tuning loop is essentially a lightweight, fully differentiable coordinate reconstruction step. In our profiling, the side-chain reconstruction accounts for only **≈1.21%** of the total wall-clock time per training step, which is negligible compared to the IgGM backbone diffusion and the TopoScorer forward/backward passes.
>
> We also note that our method is robust to moderate side-chain positioning noise. As reported in the noise-sensitivity analysis presented to another reviewer, adding coordinate jitter of 0.1–0.5 Å changes TopoScorer’s predictions by less than 5% on average, and even 1.0 Å perturbations induce only ≈10% variation. We present this additional experiment in Appendix 8.
>
> **Q5: 4.3 and Fig 2: some additional justifications of the analysis are needed: how does the pattern of $\lambda_{sum}$ indicate the physiochemical properties of the amino acids and atoms a priori?**
>
> We appreciate the reviewer’s request for a clearer justification. We would like to clarify that $λ_{sum}$  is only one of six Laplacian spectral statistics used in TopoScorer; each of these six statistics has a different and complementary topological interpretation, which we have explained in detail in Appendix A.4. In Sec. 4.3 and Fig. 2, we focus on $λ_{sum}$ because, according to the feature-importance heatmap in Fig. 2(b), $λ_{sum}$  is the **most influential** spectral feature for the considered analysis. This makes it the most illustrative example for visualizing how PTHL-based spectra reflect physicochemical patterns. In our setting, $λ_{sum}$  is defined as the sum of the non-zero eigenvalues of the (hypergraph) Laplacian within a given element- or residue-specific PTHL channel. For a standard (weighted) graph Laplacian L, this quantity equals the trace of L, which in turn is proportional to the total weighted degree / interaction strength in that channel. Intuitively, a larger $λ_{sum}$  indicates a denser, more strongly connected local interaction network, whereas a smaller $λ_{sum}$  corresponds to a sparser or more weakly interacting environment. Because our PTHL channels are physicochemically meaningful (e.g., hydrophobic contacts, polar contacts, charged interactions, backbone vs. side-chain contacts, etc.), the typical interaction patterns of different amino acids and atom types naturally induce different $λ_{sum}$  regimes a priori, before any supervised training. For example, Hydrophobic residues and carbon-rich atoms tend to cluster in the protein core and form dense, multi-body van der Waals networks, which result in systematically higher $λ_{sum}$  in hydrophobic channels; weakly interacting or geometrically isolated groups (e.g., flexible surface loops, solvent-exposed side chains) show consistently lower $λ_{sum}$ , reflecting their sparse contact environment. In this way, physicochemical preferences shape contact patterns, contact patterns determine Laplacian spectra, and the resulting $λ_{sum}$  statistics reveal these differences even without labels.

---

> ### Author Response · Authors · 2025-11-22
>
> **W1: The predictive performance is not significantly improved compared to the baselines**
>
> We thank the reviewer for carefully evaluating our model from multiple perspectives, and we are pleased to present our improved results. Model performance is a central aspect that we carefully monitor and continuously seek to improve. Since the initial submission, we have substantially strengthened the training and evaluation pipeline, including (i) scaling up and re-curating the training data, (ii) a more thorough hyperparameter search (learning rate, optimizer, weight decay, batch size, PTHL temperature parameters, radius grid, etc.), (iii) clustering highly similar interactions in the training set to reduce redundancy and stabilize optimization.
>
> We initially constructed all splits purely based on release time, which, as pointed out by Bushuiev et al. [1], can still allow substantial leakage when used as the only criterion. To address this, we revisited all our splits and adopted the interface-similarity protocol recommended in Bushuiev et al. [1,2]. We removed these near-duplicate test entries (i.e., potential leaks) and re-evaluated all baselines and TopoScorer on the resulting leakage-controlled benchmark. Importantly, after removing all potential near-duplicate interfaces, our model still **achieves** **state-of-the-art performance in** **both** **Spearman and Pearson** on  **binding affinity prediction**, as well as **SOTA Spearman correlations and second best Pearson on single-mutation and multi-mutation subsets.** The updated results are reported in the revised Table 1. For transparency and reproducibility, we provide the detailed train–test interface similarity statistics and the leakage-filtered benchmark files (including updated performance tables for all methods) in our anonymous GitHub repository linked in the manuscript.
>
> | Method       | Affinity Spearman | Affinity Pearson | Single-mutation Spearman | Single-mutation Pearson | Multi-mutations Spearman | Multi-mutations Pearson |
> |-------------|-------------------|------------------|--------------------------|-------------------------|--------------------------|-------------------------|
> | PyRosetta   | 0.1856            | 0.1954           | 0.3422                   | 0.3285                  | 0.2927                   | 0.2258                  |
> | FoldX       | *0.3295*     | 0.3008    | 0.4355                   | 0.4586                  | 0.3734                   | 0.3241                  |
> | ESM-1v      | 0.1034            | 0.0876           | 0.1524                   | 0.1921                  | 0.1512                   | 0.1736                  |
> | ESM-IF      | 0.0530            | 0.0244           | 0.1116                   | 0.1047                  | 0.1697                   | 0.0700                  |
> | PRODIGY     | 0.1549            | 0.1277           | 0.3233                   | 0.2902                  | 0.3421                   | 0.3236                  |
> | DSMBind     | 0.3072            | *0.3269*           | 0.3530                   | 0.3261                  | 0.3673                   | 0.2954                  |
> | DDGPred     | —                 | —                | *0.5522*          | 0.5303                  | 0.4585                   | 0.5638                  |
> | RDE-Network | —                 | —                | 0.5127                   | **0.6067**              | 0.5397                   | **0.6108**              |
> | GearBind    | —                 | —                | 0.5014                   | 0.5496                  | *0.5470*          | 0.5616                  |
> | TopoNetTree | —                 | —                | 0.5185                   | 0.5508                  | —                        | —                       |
> | TopoScorer  | **0.3848**        | **0.3804**       | **0.5876**               | *0.5615*          | **0.5704**               | *0.5652*           |
>
> [1] Bushuiev et al, 2024, “Revealing data leakage in protein interaction benchmarks”, https://arxiv.org/abs/2404.10457
>
> [2] Bushuiev et al., 2024, Learning to design protein–protein interactions with enhanced generalization, ICLR 2024.

---

> ### Author Response · Authors · 2025-11-22
>
> **Q3:Besides comparing with a totally different generative framework, how does classifier guidance with TopoScorer compare to other prediction or scoring models (eg compared in Table 1) as classifier guidance on the same generative model, in terms of performance, parameter size and running time?**
>
> We thank the reviewer for this suggestion and for encouraging us to evaluate TopoScorer more broadly as a guidance signal. We are currently integrating TopoScorer as a classifier-guidance term at sampling time in IgGM. Concretely, if $x_t$ denotes the current structure/sequence state and $f_\theta(x_t)$ the IgGM update, we add a TopoScorer-based guidance term
> \begin{equation}
>     x_{t+1} = x_t + \eta\bigl(f_\theta(x_t) + \lambda\\nabla_{x} S_{\text{Topo}}(x_t)\bigr),
> \end{equation}
> which is equivalent to the standard classifier-guidance decomposition
> \begin{equation}
>     \nabla_x \log p(x \mid c) = \nabla_x \log p(x) + \lambda\\nabla_x \log p(c \mid x),
> \end{equation}
> with $S_{\text{Topo}}(x)$ playing the role of a differentiable affinity score $\log p(c \mid x)$. As shown in the table below, **frozen classifier guidance yields improvements smaller than TopoScorer-based finetuning**, because the underlying IgGM parameters are not adapted to the new reward, which is also consistent with our expectation.
> | Model      | RMSD(CA) aligned | TMscore | DockQ  | SR     |
> |----------- |------------------|--------:|-------:|-------:|
> | IgGM       | 2.337086         | 0.684455 | 0.1451 | 20.77% |
> | IgGM_Topo  | **2.1807**       | **0.699** | **0.1754** | **27.26%** |
> | IgGM_guide | 2.3152           | 0.6894  | 0.1618 | 22.98% |
>
>
> Extending this comparison to other scorer models in Table 1 is non-trivial. First, many competitive strong models are \(\Delta\Delta G\)-only and not defined as absolute affinity scorers: this includes DDGPred, RDE-Network, GearBind, and TopoNetTree, which require a wild-type/mutant pair and are not directly applicable as a scalar reward. Second, several remaining baselines are either **external tools** or **lack of available gradients**: PyRosetta, FoldX, and PRODIGY are standalone tools with heavy per-sample runtime, making step-wise guidance inside a multi-step generative loop computationally prohibitive, and they do not expose gradients with respect to atomic coordinates. Sequence-based models such as ESM-1v likewise lack a natural differentiable interface to IgGM’s 3D outputs.
>
> DSMBind, while closer in spirit, is also not a drop-in classifier-guidance scorer, and using DSMBind for classifier guidance in IgGM would require substantial additional engineering (environment compatibility, matching noise schedules, wrapping its score into a calibrated scalar reward). For these reasons, we focus our classifier-guidance implementation on TopoScorer itself, and we believe that the results from our binding‐affinity prediction benchmarks, together with the IgGM finetuning experiments, already provide an indirect but strong indication of TopoScorer’s suitability for classifier guidance.

---

> > ### Author Response · Authors · 2025-11-27
> >
> > Dear Reviewer,
> >
> > Happy Thanksgiving! We sincerely appreciate your time and efforts in reviewing our manuscript and offering constructive suggestions. As the discussion phase is approaching its end, we would like to kindly confirm whether we have sufficiently addressed your concerns. Should there be any remaining questions requiring further clarification, please do not hesitate to let us know. If you are satisfied with our responses, we would greatly appreciate your consideration in adjusting the evaluation scores accordingly.
> >
> > We look forward to your feedback.
> >
> > Best regards,
> >
> > Anonymous Authors

---

> > > ### Comment · Reviewer_dETA · 2025-11-28
> > >
> > > I appreciate the authors' detailed responses. I would be happy to raise the score to positive in the final ratings.

---

> > > > ### Author Response · Authors · 2025-11-28
> > > >
> > > > Thank you very much for your thoughtful comments and for your positive reassessment of our work. We truly appreciate your recognition and your willingness to raise the score in the final ratings.

---

### Meta-Review · Area_Chair_HXGy · 2026-01-06

**Summary:**

This paper proposes TopoScorer, a parameter-efficient model for predicting protein–protein interaction affinities based on topological features of complex structures. The model represents PPI complexes using multi-channel graphs that encode atom types, physicochemical properties, and role information, and is trained end-to-end for affinity prediction and mutation effect prediction. Experimental results show that TopoScorer achieves performance comparable to existing state-of-the-art methods on the evaluated benchmarks.

**Reviewer Concerns:**

Three reviewers initially rated the paper below the acceptance threshold. The main concerns were that the performance does not show a clear improvement over prior approaches. Reviewers also raised questions about the robustness of the model to structural noise, as well as whether the claims of being lightweight, interpretable, and end-to-end trainable were sufficiently supported by evidence.

In response, the authors provided a careful rebuttal. They added inference time comparisons and parameter count analyses, demonstrating clear advantages in both computational efficiency and model size relative to competing methods.

Despite these improvements, some key issues remain unresolved. From my perspective, the paper is borderline. The methodological novelty is limited, and the performance gains over earlier methods, such as TopoNetTree (2020), are relatively modest. Given the small size of the test sets, these improvements are not entirely convincing. In addition, while robustness to structural noise is emphasized, many existing approaches (including DSMBind) rely heavily on high-quality crystal structures. The authors report that the average relative difference in predicted affinity between AF3-predicted structures and crystal structures is only 0.13%, which is unexpectedly small. This result warrants careful verification and should be clearly reported in the main text. If validated, it would represent a meaningful practical contribution, as crystal structures are rarely available in real-world applications.

Overall, the work is interesting but not yet mature, and additional experimental validation would be necessary to convincingly demonstrate its advantages in performance and robustness.

**Reviewer Scores:**

I do not expect any reviewer scores to change a lot.

---

### Decision · Program_Chairs · 2026-01-26

Reject